# Analysis of single-Alter shielded and unshielded measurements of mixed and solid precipitation from WMO-SPICE

John Kochendorfer[1], Rodica Nitu[2,3], Mareile Wolff[4], Eva Mekis[2], Roy Rasmussen[5], Bruce Baker[1], Michael E. Earle[6], Audrey Reverdin[7], Kai Wong[2], Craig D. Smith[8], Daqing Yang[8], Yves-Alain Roulet[7], Samuel Buisan[9], Timo Laine[10], Gyuwon Lee[11], Jose Luis C. Aceituno[9], Javier Alastrué[9], Ketil Isaksen[4], Tilden Meyers[1], Ragnar Brækkan[4], Scott Landolt[5], Al Jachcik[5], and Antti Poikonen[10]

[1]Atmospheric Turbulence and Diffusion Division, ARL, National Oceanic and Atmospheric Administration, Oak Ridge, TN, 37830, US
[2]Environment and Climate Change Canada, Toronto, ON, M3H 5T4, Canada
[3]World Meteorological Organization, Geneva, CH-1211, Switzerland
[4]Norwegian Meteorological Institute, Oslo, 0313, Norway
[5]National Center for Atmospheric Research, Boulder, 80305, US
[6]Environment and Climate Change Canada, Dartmouth, Nova Scotia, B2Y 2N6, Canada
[7]Meteoswiss, Payerne, CH-1530, Switzerland
[8]Environment and Climate Change Canada, Saskatoon, SK S7N 3H5, Canada
[9]Delegación Territorial de AEMET (Spanish National Meteorological Agency) en Aragón, Zaragoza, 50007 Spain
[10]Finnish Meteorological Institute, Helsinki, FI-00101, Finland
[11]Kyungpook National University, Daegu, 41566, Korea

*Correspondence to*: John Kochendorfer (john.kochendorfer@noaa.gov)

**Abstract.** Although precipitation has been measured for many centuries, precipitation measurements are still beset with significant inaccuracies. Solid precipitation is particularly difficult to measure accurately, and winter-time precipitation measurement biases between different measurement networks or different regions can exceed 100%. Using precipitation gauge results from the World Meteorological Organization Solid Precipitation Intercomparison Experiment (WMO-SPICE), errors in precipitation measurement caused by gauge uncertainty, spatial variability in precipitation, hydrometeor type, crystal habit, and wind were quantified. The methods used to calculate gauge catch efficiency and correct known biases are described. Adjustments, in the form of 'transfer functions' that describe catch efficiency as a function of air temperature and wind speed, were derived using measurements from eight separate WMO-SPICE sites for both unshielded and single-Alter shielded weighing precipitation gauges. For the unshielded gauges, the average undercatch for all eight sites was 0.50 mm hr$^{-1}$ (34%), and for the single-Alter shielded gauges it was 0.35 mm hr$^{-1}$ (24%). After adjustment, the mean bias for both the unshielded and single-Alter measurements was within 0.03 mm hr$^{-1}$ (2%) of zero. The use of multiple sites to derive such adjustments makes these results unique and more broadly applicable to other sites with various climatic conditions. In addition, errors associated with the use of a single transfer function to correct gauge undercatch at multiple sites were estimated.

## 1 Introduction

Like many atmospheric measurements, precipitation is subject to the observer effect, whereby the act of observing affects the observation itself. Hydrometeors falling towards a precipitation gauge can be deflected away from the gauge inlet due to changes in the speed and direction of the airflow around the gauge that are caused by the gauge itself (eg. Sevruk et al., 1991). The magnitude of this effect varies with wind speed, wind shielding, the shape of the precipitation gauge, and the predominant size, phase, and fall velocity of the hydrometeors (Colli et al., 2015; Folland, 1988; Groisman et al., 1991; Theriault et al., 2012; Wolff et al., 2013). Because all these factors affect the amount of undercatch, it is difficult to accurately describe and adjust the resultant errors for all gauges, in all places, in all types of weather. This has been an active area of research for over 100 years (eg. Alter, 1937; Heberden, 1769; Jevons, 1861; Nipher, 1878), with significant findings for manual measurements described in the WMO precipitation intercomparison experiment performed in the 1990s (Goodison et al., 1998; Yang et al., 1998b; Yang et al., 1995). More recently, studies of the magnitude and importance of such measurement errors have been performed using both analytical (Colli et al., 2016; Colli et al., 2015; Nespor and Sevruk, 1999; Theriault et al., 2012) and observational approaches (eg. Chen et al., 2015; Ma et al., 2015; Rasmussen et al., 2012; Wolff et al., 2013; Wolff et al., 2015). The ultimate goal of all this research is to facilitate the creation of accurate and consistent precipitation records spanning different climates and different measurement networks, including measurements by different gauges and shields (eg. Førland and Hanssen-Bauer, 2000; Scaff et al., 2015; Yang and Ohata, 2001).

Due to the importance of precipitation measurements for hydrological, climate, and weather research, and also due to the many unanswered questions and uncertainties that are associated in particular with solid precipitation (Førland and Hanssen-Bauer, 2000; Goodison et al., 1998; Sevruk et al., 2009), the WMO began planning an international intercomparison focused on solid precipitation in 2010. The primary goals of this intercomparison include the assessment of new automated gauges and wind shields, and the development of adjustments for these gauges and wind shields.

Results from the WMO Solid Precipitation Intercomparison Experiment (WMO-SPICE) were used to develop adjustments for different types of weighing gauges, within different types of shields. Due to the nature of this unique dataset, including periods of precipitation from many different sites, gauges and shields, new analysis techniques were used to develop adjustments and quantify the errors inherent in applying such corrections. The focus of the work described below is on the unshielded and single-Alter-shielded weighing precipitation gauges. Based on results of a previous CIMO survey (Nitu and Wong, 2010), WMO-SPICE selected two of the most ubiquitous configurations used in national networks for the measurement of solid precipitation. As requested by WMO-SPICE, both unshielded and single-Alter shielded weighing gauges were present at all of the WMO-SPICE testbeds. Eight of these sites also had a Double Fence Automated Reference (DFAR) configuration (Nitu, 2012), which is essentially an automated weighing gauge within a Double Fence Intercomparison Reference (DFIR). The DFIR is described in detail in the previous WMO-Solid Precipitation

Intercomparison report (Goodison et al., 1998). The measurements from these gauges provided a unique opportunity to develop and test wind corrections for unshielded and single Alter shielded gauges at multiple sites.

A testbed with well-shielded gauge installed either within well-maintained bushes (Yang, 2014) or large, octagonal, concentric wooden fences (Golubev, 1989, 1986; Yang et al., 1998b) is required to develop a precipitation transfer function. Such transfer functions are typically developed at one site and used to adjust precipitation measurements recorded elsewhere. This raises the obvious question of how universal such adjustments are. At a given site, the catch efficiency (*CE*) of an unshielded or a single-Alter shielded gauge can vary significantly (eg. Ma et al., 2015; Rasmussen et al., 2012; Theriault et al., 2012; Wolff et al., 2015). Some of this variability is driven by differences in ice crystal shape (habit), mean hydrometeor fall velocity, and hydrometeor size (eg. Theriault et al., 2012), so it follows that there may be significant variability in catch efficiency from site to site. In addition, issues such as complex topography may contribute to the site-specific variability in catch efficiency. The magnitude of this site-to-site variability has not been previously quantified.

In the present study, measurements from eight different WMO-SPICE testbeds, varying significantly in their siting, elevation, and climate, allowed us to address the important and long-standing question of how applicable a single transfer function is for multiple sites. Measurements from all sites, spanning two winter seasons from 2013-2015, were combined to develop robust multi-site transfer functions, marking a significant departure and improvement over past work, in which typically only one (Folland, 1988; Smith, 2009; Wolff et al., 2015) or sometimes two (Kochendorfer et al., 2017) sites were used to develop transfer functions. In addition, the resultant multi-site or 'universal' transfer functions were tested on each site individually, revealing the magnitude of site-specific errors and biases for the different types of transfer functions developed.

## 2 Methods

### 2.1 Precipitation measurements

Precipitation measurements from eight sites were included in this analysis, each of which had a DFAR configuration. They include the Canadian CARE site (CARE), the Norwegian Haukeliseter site (Hauk), the Swiss Weissfluhjoch site (Weis), the Finnish Sodankylä site (Sod), the Canadian Caribou Creek site (CaCr), the Spanish Formigal site (For), the US Marshall site (Ma), and the Canadian Bratt's Lake site (BrLa). Site locations are shown in Fig. 1. These sites are described in detail in the WMO-SPICE commissioning reports (available here: http://www.wmo.int/pages/prog/www/IMOP/intercomparisons/SPICE/SPICE.html). Basic background information such as site elevation, and air temperature and wind speed statistics during precipitation are included in Table 1. Each of these sites also had the tertiary WMO-SPICE precipitation reference installation, which consisted of both an unshielded and a single-Alter shielded weighing precipitation gauge. This tertiary reference was proposed for sites that could not support the

installation of a large DFAR, and it was also included at all of the DFAR sites for evaluation. As a result, these eight sites each had three 'reference' weighing gauges, one within a DFIR shield (the DFAR), one within a single-Alter shield, and one unshielded (Nitu, 2012). The results presented here are derived from these measurements, but only the DFAR was treated as a reference.

Except for Formigal, Spain, where DFAR measurements were only available during the winter of 2014-2015, measurements from the winter seasons (Oct 1 – April 30) of 2013-2014 and 2014-2015 were used in this analysis. For WMO-SPICE, the reference weighing precipitation gauges were designated by the International Organizing Committee as the OTT Pluvio$^2$ (OTT Hydromet, Kempten, Germany) and the Geonor T-200B3 (3-wire T200B, Geonor Inc., Oslo, Norway). All of the precipitation measurements used in this analysis were recorded using these host-provided, 'reference' weighing gauges. Of the sites considered in this analysis, three used the Pluvio$^2$ gauge (Sodankylä, Weissfluhjoch, and Formigal), and the other five used the Geonor T-200B3 gauge (CARE, Haukeliseter, Caribou Creek, Marshall, and Bratt's Lake). Some sites had both types of gauges, with preliminary analysis indicating no significant differences between the Pluvio$^2$ and Geonor T-200B3 DFAR (Ryu et al., 2016). Fig. 2 also demonstrates the agreement between the unshielded Pluvio$^2$ and Geonor T-200B3 measurements at CARE. A set of 389 single-Alter shielded Pluvio$^2$ and Geonor measurements were available for comparison at CARE, resulting in a slope of 1.01, offset of -0.002, and a root mean square error (RMSE) of 0.085 mm. Based on these findings, the Pluvio$^2$ and Geonor T-200B3 measurements were treated as equivalent, with the analysis methods developed here (and for the entire WMO-SPICE experiment) based on the hypothesis that the effects of shielding, siting, and climate were more important than the type of reference weighing precipitation gauge used.

## 2.2 Analysis methods

### 2.2.1 Data quality control

Data from each site were processed using a standardized quality control procedure. The procedure was developed and tested using WMO-SPICE data and consisted of the following steps:

1) A format check, to ensure that the data had the correct number of records per day, as expected based on the instrument sampling resolution (data with repeated time stamps were removed, and any missing time stamps were filled with 'null' values);

2) A range check, to identify and remove values that were outside of the manufacturer-specified output range for each sensor;

3) A jump filter, to identify and remove points exceeding the maximum point-to-point variation expected for a given sensor;

4) A smoothing step, employing a Gaussian filter to mitigate the influence of high-frequency noise on instrument data (applied only to data from weighing gauges).

The above steps were applied uniformly at a central archive to the one minute measurements, with thresholds and filter parameters defined separately for each instrument and sampling resolution. Data were also flagged according to the results of the above checks and filters. Data and flags were output at 1 minute temporal resolution. In addition a manual assessment was used to identify and account for any periods during which instrument data were not available (e.g. instrument

maintenance, site power outage) or during which instrument performance may have been compromised (e.g. frozen sensors). These steps are described in more detail in Reverdin (2016).

### 2.2.2 Selection of 30 minutes precipitation events

To ensure that the datasets used for the development of transfer functions were as consistent as possible across all sites, and represented precipitating periods with a high degree of confidence, a methodology was established to identify precipitation

events for further analysis using the 1 minute, quality-controlled data. The occurrence of a precipitation event was predicated on the fulfilment of two criteria: first, accumulated precipitation $\geq 0.25$ mm over 30 minutes reported by the weighing gauge in the DFAR; and second, the occurrence of precipitation over at least 60% of the same 30 minute interval by a sensitive precipitation detector (e.g. disdrometers or optical sensors). The precipitation detector measurements were included to provide independent verification of the occurrence of precipitation and help accurately identify periods of precipitation.

The rationale behind the use of 30 minute intervals and a 0.25 mm accumulation threshold for precipitation events is detailed elsewhere (Kochendorfer et al., 2017). To summarize this rational, the 0.25 mm threshold was found to reduce the effects of measurement noise on the selection of events, while the 30 minute interval provided a large sample size of events. The 60% threshold for precipitation occurrence – corresponding to 18 minutes within a 30 minute interval – was chosen primarily to

ensure that the accumulation reported by the weighing gauge in the DFAR represented falling precipitation, and not one or more 'dumps' of precipitation accumulated on the gauge housing/orifice into the bucket, or any other type of false accumulation. Separate tests, which are not described in detail here, indicated that the number of 30-min events selected was insensitive to the specific value of the threshold, provided it was between 50% and 80%.

The event selection procedure was applied uniformly to the 1 minute quality-controlled datasets from each site. The accumulation reported by the weighing gauge in the DFAR and each single-Alter shielded and unshielded weighing gauge was determined for each event. Ancillary parameters, such as ambient temperature (minimum, maximum, and mean) and mean wind speed, were also determined for each event. Flags created in the quality control process were also aggregated and reported for each event. The resultant Site Event Datasets (SEDS), which included all 30 minute precipitation events selected

within a winter season for a specific site, provided the basis for the development of these transfer functions.

### 2.2.3 Filtering of precipitation events

#### 2.2.3.1 Wind speed and direction

Additional filtering was performed on the resultant 30-min SEDS for the transfer function development. At several sites, data from wind directions associated with wind-shadowing from towers, shields, buildings, and other obstructions were removed from the record. Single-Alter gauge measurements at CARE were affected by a neighbouring windshield for wind directions within ± 45° of due south. Single-Alter and unshielded measurements at Haukeliseter were excluded for wind directions greater than 290° and less than 120° due to interference from the DFAR. At Weissfluhjoch, wind measurements associated with wind directions that were either greater than 340° or less than 200° were excluded from analysis due to wind shadowing caused by the DFAR and a building. For Bratt's Lake, 30-min precipitation measurements with a 10 m height wind speed greater than 10 m s$^{-1}$ were excluded from the analysis, as this site was prone to blowing snow events at high wind speeds.

Unrealistic wind speed ($U$) measurements were also removed as described below, with mean 30 minute air temperature ($T_{air}$) and wind speed measurements used for all of the analysis presented here. At all sites, wind speeds equal to zero were removed. At CARE, gauge height wind speeds that were greater than the 10 m height wind speeds were removed, as they appeared to be the result of the 10 m propeller anemometer being partially frozen. Similar behaviour was observed with the 10 m propeller anemometer at Marshall in measurements pre-dating SPICE; in an isolated event, the propeller anemometer recorded non-zero wind speeds that were much lower than the gauge height wind speeds due to the effects of ice on the anemometer. At Haukeliseter, 'stuck' wind speeds – observations with no change from one 30 minute period to the next - were removed. At Marshall, the sonic anemometer measurements at a height of 10 m, which was one of two anemometers used to determine the 10 m wind speed, reported erratic measurements below 0.9 m s$^{-1}$ and were removed. Following Kochendorfer et al. (2017), the 10 m sonic anemometer measurements at Marshall were used only when the propeller anemometer measurements were affected by the wind shadow of the sonic anemometer.

#### 2.2.3.2 Accumulation thresholds for gauges under test

Minimum accumulation thresholds were also used for all the gauges under test (UT), to account for random variability in the event accumulation values. The use of a minimum threshold was necessary because even the DFAR precipitation measurements were subject to random variability. Tests performed using identical gauge-shield combinations revealed that the application of a minimum threshold to only one gauge arbitrarily included some events near the threshold and excluded others, and thereby biased the results towards the gauge used for the event selection. Following Kochendorfer et al. (2017), the minimum threshold for the gauge under test was estimated using Eq. 1.

$$THOLD_{UT} = median \left( \frac{P_{UT}}{P_{DFAR}} \right) \times 0.25 \; mm \tag{1}$$

where $THOLD_{UT}$ is the threshold accumulation for the gauge under test, $P_{UT}$ is the 30-min accumulated precipitation from the gauge under test, and $P_{DFAR}$ is the 30-min DFAR reference accumulation. Because the results were sensitive to the magnitude of the threshold for the gauge under test, this study used a conservative approach that included only solid precipitation measurements (mean $T_{air}$ < -2 °C) with relatively high winds (5 m s$^{-1}$ < $U_{10m}$ < 9 m s$^{-1}$) in the determination of the $THOLD_{UT}$ (Eq.1). When all available measurements were used for the determination of the minimum threshold for the gauge under test, the inclusion of rain and low wind speed measurements resulted in a higher minimum threshold, erroneously excluding low-rate, low-catch-efficiency solid precipitation measurements from the analysis.

A minimum threshold was calculated for the unshielded gauges ($THOLD_{UN}$ = 0.06 mm/30 min) and for the single-Alter shielded gauges ($THOLD_{SA}$ = 0.11 mm/30 min), and all events with 30-min accumulation values below the respective thresholds were excluded from the analysis. Unrealistically large accumulations were also removed using the measured catch efficiency ($CE = P_{UT}/P_{DFAR}$), with large outliers from the gauge under test identified and excluded from the analysis when the catch efficiency was more than three standard deviations greater than 1.0: $CE > [3 \times stdev(CE) + 1.0]$. The resultant maximum unshielded (UN) threshold for the 30-min measurements was 1.93 x $P_{DFAR}$, and the maximum single-Alter (SA) threshold was 1.98 x $P_{DFAR}$.

### 2.2.4 Wind speed estimation

To develop transfer functions for both 10 m and gauge-height wind speeds, the best available wind speed sensors at every site were used to estimate the 10 m and gauge-height wind speeds. Because the availability and quality of wind speed measurements varied from site to site, the methods used for wind speed estimation also varied. Generally, the log-profile law was applied to estimate the change in wind speed with height, assuming neutral surface layer stability.

$$U_z \propto \ln\left[ (z - d)/z_0 \right] \tag{2}$$

where $U_z$ is the wind speed ($U$) at a height $z$, $z_0$ is the roughness length, and $d$ is the displacement height. Using 30-min mean wind speeds uncompromised by obstacles, the roughness length and displacement height were estimated for sites with wind profile measurements (Thom, 1975). For sites without wind speed measurements at multiple heights, a generic roughness length ($z_0$ = 0.01 m) and displacement height ($d$ = 0.4 m) were used, as these values were fairly representative of the sites with wind speed profile measurements available. At CARE, the actual gauge height and 10 m height wind speeds were used. At Formigal, Marshall and Haukeliseter, the 10 m wind speed measurements were used to estimate the gauge height wind speeds. At Marshall and Haukeliseter the roughness length and displacement height were determined using wind speed profile measurements available from unobstructed wind directions. At Formigal there was no gauge height wind speed sensor available, so the generic roughness length and displacement height were used to estimate the gauge height wind speed. At Weissfluhjoch, Sodankylä, Caribou Creek, and Bratt's Lake, there were no 10 m height wind speed measurements

available, and the gauge height wind speed was used to estimate the 10 m height wind speed using the generic roughness length and displacement height.

### 2.2.5 Selection of light precipitation events

In addition to the selection of events used to create the transfer functions, light precipitation events (accumulated DFAR precipitation < 0.25 mm) were also selected for several sites. These Site Light Event Datasets (SLEDS) were used as independent measurements available for the validation of the transfer functions, and also to assess the utility of the transfer functions for the correction of light precipitation events. This was also motivated by the importance and challenge of measuring precipitation in Polar Regions, where a significant amount of the annual precipitation occurs as very light snow (Mekis, 2005; Mekis and Vincent, 2011; Metcalfe et al., 1994; Yang et al., 1998a).

For the light precipitation analysis, four sites were selected: CARE, Bratt's Lake, Haukeliseter, and Sodankylä (Fig. 1). For automated gauges, the natural fluctuation (noise) around zero can easily be confused with light precipitation. To help distinguish between noise and precipitation, a minimum threshold of 0.1 mm in 30 min was chosen for the reference gauge. In addition, independent verification of precipitation occurrence was provided by a sensitive precipitation detector, following the same methodology used for the SEDS (Section 2.2.2). Light events were selected only when the precipitation detector observed precipitation for at least 18 min of the 30 minute period.

The above criteria were applied to the quality-controlled, 1 minute datasets from the selected sites. The selected events were filtered further using the procedures outlined in Section 2.2.3. For the tested unshielded / single-Alter shielded gauges, minimum threshold values equivalent to those used for the SEDS were estimated by replacing 0.25 mm in Eq. 1 with 0.1 mm. The resultant unshielded / single-Alter shielded 0.025 / 0.043 mm minimum threshold values were applied, with all 30-min periods with less than the minimum threshold excluded from the analysis. Where possible, the gauge height wind speed was used in the computation.

### 2.2.6 Selection of 12 and 24 hour precipitation events

Because many precipitation measurements are only available over longer time periods, 12 and 24 h precipitation accumulations were also used for the testing and development of transfer functions. These longer precipitation accumulations were created using a minimum threshold of 1.0 mm per each 12 or 24 h period as measured by the DFAR, and a minimum of 15 min of precipitation as measured by the precipitation detector. Minimum and maximum thresholds for the unshielded and single-Alter shielded measurements were calculated and applied using the same methods that were applied to the 30-min precipitation measurements, described in Section 2.2.3.2. Accompanying mean air temperatures and wind speeds were also calculated for the 12 and 24 h precipitation accumulations.

### 2.2.7 Transfer function models

A single transfer function of $T_{air}$ and $U$ was created using all the similarly-shielded/unshielded precipitation gauge measurements. For example, the unshielded precipitation measurements from all eight sites were grouped together irrespective of whether they were recorded using a Pluvio[2] or a Geonor T-200B3 gauge, and a transfer function was determined using these measurements and the corresponding DFAR measurements. Equation 3 describes the form of the transfer function used, as introduced by Kochendorfer et al. (2017):

$$CE = e^{-a(U)(1-\tan^{-1}(b(T_{air}))+c)} \tag{3}$$

where $U$ is wind speed, $T_{air}$ is the air temperature, and $a$, $b$, and $c$ are coefficients fit to the data. The sigmoid transfer function (Wolff et al., 2013) was also tested with these data, but like in Kochendorfer et al. (2017), the more complex sigmoid function produced similar biases and RMSE, so the simpler Eq. 3 was used.

Without explicitly including $T_{air}$, transfer functions for mixed and solid precipitation were also created separately as an exponential function of wind speed:

$$CE = (a)e^{-b(U)} + c \tag{4}$$

where $a$, $b$, and $c$ are coefficients fit to the data. This was done for comparison with past studies that used similar techniques (eg. Goodison, 1978; Yang et al., 1998b), to make simple adjustments available to users, and also to help quantify the advantages and disadvantages of explicitly including $T_{air}$ in transfer functions.

Due to the prevalence of air temperature measurements in observing networks, and the fact that not all of the WMO-SPICE sites included precipitation type measurements, the 30-min mean $T_{air}$ was used to determine precipitation type for Eq. 4. Solid precipitation was defined as $T_{air} <$ -2 °C, and mixed precipitation was defined as 2 °C $\geq T_{air} \geq$ -2 °C (Kochendorfer et al., 2017; Wolff et al., 2015). Liquid precipitation ($T_{air} >$ 2 °C) data were also evaluated, but due to the limited quantity of warm-season measurements in this dataset and the negligible magnitude of the liquid precipitation adjustment derived from these measurements, no Eq. 4 type transfer functions were created for liquid precipitation. For comparison with the Eq. 3 results, the resultant transfer functions were used to adjust the precipitation measurements, with no correction applied to the liquid precipitation measurements for the adjusted Eq. 4 transfer function results.

### 2.2.8 Wind speed threshold

The resultant transfer functions were only valid for the range of wind speeds available in the measurements from which they were derived. At high mean wind speeds, where precipitation measurements were scarce or non-existent, the transfer functions were unconstrained and inaccurate. Therefore a wind speed threshold was required, above which the resultant transfer functions cannot be used. This wind speed threshold was chosen by assessing the availability of high wind speed results for all air temperatures after plotting catch efficiency in three-dimensions as a function of air temperature and wind

speed. The mean wind speed threshold at gauge height was 7.2 m s$^{-1}$, and at 10 m it was 9 m s$^{-1}$. Following our recommendation for the application of these transfer functions, for the transfer function validation mean wind speeds exceeding the wind speed threshold were replaced by the value of the wind speed threshold.

### 2.2.9 Transfer function testing

Transfer functions were applied to obtain the adjusted SA and UN precipitation measurements at each site. Statistics were then calculated for each site by comparing the adjusted SA or UN gauge precipitation to the DFAR precipitation. This approach was chosen because it produced more universal multi-site transfer functions, with a single transfer function describing all of the available sites within WMO-SPICE, while simultaneously providing realistic estimates of the magnitudes of site biases that occurred due to local variations in climate and siting. Transfer function statistics were also

produced for the entire dataset by combining the adjusted SA/UN measurements from all of the sites together and comparing them to the corresponding DFAR measurements.

Four different statistics were used to quantify errors in the different types of measurements and adjustments. These statistics were all based on the 30-min precipitation measurements. The root mean square error (*RMSE*) was calculated based on the

difference between the measurements under test and the corresponding DFAR measurements. The mean bias was also calculated from the difference between the mean of the precipitation measurements under test and the mean of the corresponding DFAR precipitation measurements. The correlation coefficient (*r*) between the measurements under test and the DFAR was calculated. In addition, the Percentage of Events (*PE*) was introduced in this study as a means of assessing the frequency with which the test and reference measurements agreed within a specified threshold. The *PE* is defined as the

ratio of the number of events within the threshold to the total number of events, expressed as a percentage. For all but the light events, the threshold was 0.1 mm; for the light events, the threshold was decreased to 0.05 mm to achieve higher sensitivity. The *PE* statistic was included because it is less sensitive to individual outliers, and provided an alternative assessment of the overall performance of the adjustments.

To test the appropriateness of the transfer functions for the adjustment of 12 and 24 h precipitation records, the 30-min transfer functions were applied to 12 and 24 h precipitation measurements. To better evaluate the effects of using these longer time periods, new Eq. 3 type transfer functions were also derived using the 12 and 24 h measurements and the same methods described for the 30-min events. Error statistics for the 12 and 24 h measurements were calculated and compared for both the 30-min transfer functions and the appropriate 12 and 24 h transfer functions.

## 3 Results and discussion

### 3.1 Unshielded precipitation measurements

Unshielded gauge measurements from all eight sites were used to create and test transfer functions. The resultant unshielded Eq. 3 transfer function coefficients for both gauge height and 10 m wind speeds are given in Table 2. Eq. 4 transfer coefficients were also produced for both mixed and solid precipitation (Table 3). The three-dimensional (Eq. 3) transfer functions in Fig. 3 are shown as a function of wind speed only, with $T_{air}$ set to -5 °C. This $T_{air}$ value of -5 °C was determined from the median $T_{air}$ (-5.2 °C) during periods of solid precipitation. For comparison with past work, the Eq. 3 type transfer function with the coefficients from Kochendorfer et al. (2017) is also included in Fig. 3, using the same $T_{air}$ value of -5 °C to display the transfer function on a two-dimensional plot. The Kochendorfer et al. (2017) transfer function, which included earlier unshielded measurements from Marshall (predating the WMO-SPICE measurements), was very similar to the new Eq. 3 function developed using measurements from all eight WMO-SPICE sites. In addition, the Eq. 3 and Eq. 4 transfer functions responded quite similarly to wind speed below the wind speed threshold, but differences between the Eq. 3 and Eq. 4 functions near and above the wind speed threshold were larger (Fig. 3).

As an example of the effects of the application of the transfer functions, the corrected and uncorrected unshielded measurements are compared to the DFAR measurements for CARE and Marshall in Fig. 4. The necessity of the adjustments is apparent from the uncorrected measurements (Fig. 4a and 4 c), which often show smaller accumulations than those reported by the weighing gauge in the DFAR. Both the improvement and the remaining uncertainty in the corrected measurements are demonstrated in Fig. 4 b and 4 d. Adjusted and unadjusted measurements like these were used to produce error statistics for all eight sites.

The associated RMSE, bias, correlation coefficient ($r$), and the percentage of events within 0.1 mm ($PE_{0.1\ mm}$; percentage of events with less than 0.1 mm difference between the adjusted UN accumulation and the DFAR accumulation) were estimated for all eight sites using both the Eq. 3 and the Eq. 4 transfer functions, for both the gauge height and 10 m wind speeds (Fig. 5). The unadjusted unshielded RMSE, bias, $r$, and $PE_{0.1\ mm}$ are also included in Fig. 5. The measurements were typically improved by the application of the transfer functions. As expected, the change in bias was considerable (Fig. 5 b), as all of the gauges exhibited a significant negative bias (indicating undercatch) before adjustment, and after adjustment the biases were closer to zero and more variable in sign. Likewise, the percentage of events within 0.1 mm of the DFAR reported values ($PE_{0.1\ mm}$) demonstrated consistent improvement as a result of adjusting the measurements (Fig. 5 d). The correlation and the RMSE were also typically improved by adjusting the measurements, although the differences between the unadjusted and adjusted RMSE and correlations were less significant than they were for the bias and $PE_{0.1\ mm}$.

The resultant errors were not affected significantly by the type of transfer function or wind speed measurement height used, although there were small differences between the $U_{gh}$ and $U_{10m}$ adjustments at some of the sites, such as Haukeliseter, Marshall, and Weissfluhjoch. These small differences were likely driven by differences in the way the different wind speeds were determined, as Marshall and Haukeliseter had both a 10 m wind speed and a gauge height wind speed measurement, while Weissfluhjoch had only one wind speed measurement height available. Unfortunately, such discrepancies in the available wind speed measurements make it difficult to conclude anything significant about the advantages and disadvantages of gauge height wind speed measurements using these data.

The Eq. 3 and Eq. 4 transfer functions generally produced similar results, with the exception of Weissfluhjoch, where the differences between Eq. 3 and Eq. 4 apparent in Fig. 2 at and just below the wind speed threshold resulted in more significant errors for the Eq. 3 transfer function (Fig. 5). These differences may be specific to the population of data that was available to create and test these transfer functions, rather than being indicative of a general advantage of one correction type over the other.

The most significant exception to the general success of the unshielded transfer functions was at Weissfluhjoch, where the RMSE actually increased after adjustment (Fig. 5 a). Most of the measurements at Weissfluhjoch were improved by adjustment, as indicated by the significant increase in the frequency of adjusted measurements within 0.1 mm of the reference ($PE_{0.1\ mm}$, Fig. 5 d), but for some events, the adjustments greatly increased the difference between the unshielded gauge and the DFAR. Figure 6 a illustrates how this occurred, with large errors observed in the adjusted Weissfluhjoch measurements at high wind speeds. At wind speeds above 5 m s$^{-1}$, the catch efficiency at Weissfluhjoch was generally much higher than at any of the other sites. The transfer functions developed for all the sites worked well at Weissfluhjoch at lower wind speeds, but could greatly overcorrect the unshielded gauge at higher wind speeds. This resulted in a large mean bias and RMSE. For comparison with a more typical high wind site, results from Haukeliseter are shown in Fig. 6 b. One hypothesis is that the Weissfluhjoch measurements were impacted by complex topography. It is possible that the wind speed was not homogenous throughout the site, and the measured wind speed was not representative of the wind speed at the location of the unshielded and single-Alter shielded gauges. Heterogeneous winds and/or significant mean vertical wind speeds may also have caused the DFAR to underestimate precipitation in high winds. To determine the effects of the Weissfluhjoch measurements on the derived multi-site transfer functions, a sensitivity test performed using the unshielded measurements indicated that exclusion of the Weissfluhjoch measurements did not significantly affect the resultant transfer functions.

The general trend found for the Weissfluhjoch errors was valid for all sites, with the RMSE, bias, and correlation driven by the high wind speed results. This is partially because at high wind speeds in cold, snowy conditions, the transfer function adjustment more than doubled the amount of measured precipitation. Such large adjustments could significantly enhance

errors in the adjusted catch efficiency, especially when the measured catch efficiency was higher than typical; at high wind speeds, a relatively small error in the measured precipitation is doubled or even tripled, resulting in errors in the adjusted precipitation of similar magnitude to the DFAR measurement itself. For this reason alone, errors in the adjusted catch efficiency look significantly different than errors in the measured CE. This highlights the value of determining errors and biases in the adjusted precipitation measurements rather than errors in the transfer functions. If necessary, cross-validation (e.g. Kochendorfer et al., 2017) or other statistical bootstrapping techniques can be used to independently validate transfer functions and estimate errors in transfer functions.

Another general trend in the results was that unshielded measurements from the sites with complex topography were more difficult to adjust, with the transfer functions performing worst at the mountainous sites. The average unshielded RMSE for the mountainous sites (Haukeliseter, Formigal, and Weissfluhjoch) was decreased from 0.48 mm (58.6%) to 0.43 mm (53.7%) by the adjustments, while for the other sites (CARE, Sodankylä, Caribou Creek, Marshall, and Bratt's Lake) it was decreased from 0.27 mm (42.0%) to 0.20 mm (31.5%). The mean of the absolute value of the unshielded biases for the mountainous sites was decreased from 0.33 mm (41.5%) to 0.14 mm (18.0%), and for the other sites it was decreased from 0.18 mm (28.4%) to 0.04 mm (6.5%) by the adjustments. The errors in both the adjusted and the unadjusted mountainous measurements were much larger than the errors from the other sites.

### 3.2 Single-alter shielded precipitation measurements

The single-Alter shielded measurements from all eight sites were combined and used to develop transfer functions. Table 2 describes the resultant Eq. 3 transfer functions, and Table 3 describes the Eq. 4 transfer functions. Transfer functions created using both the gauge height wind speeds and the 10 m wind speeds were produced for the single-Alter shielded measurements. The wind speed thresholds used when applying the transfer functions are shown in Tables 3 and 4. For wind speeds exceeding the threshold values, the transfer functions should be applied by forcing the actual wind speed to the wind speed threshold, as discussed in Sections 2.2.5 and 3.1.

The resultant Eq. 3 and Eq. 4 transfer functions for single-Alter measurements showed greater similarity to each other (Fig. 7) than the unshielded transfer functions (Fig. 3). They were also similar to the Kochendorfer et al. (2017) transfer functions (Fig. 7), developed using earlier measurements from only Haukeliseter and Marshall. Application of the single-Alter transfer functions reduced the RMSE at most of the sites in comparison to the unadjusted measurements (Fig. 8 a). In addition, the results were relatively insensitive to the methods used to produce the adjusted measurements. The RMSE values were quite similar using Eq. 3 and Eq. 4, and they were not affected significantly by the wind speed measurement height. Like the unshielded measurements, the biases (Fig. 8 b) and the percentage of events within 0.1 mm ($PE_{0.1\ mm}$, Fig. 8 d) were more significantly improved by the application of the transfer functions than the RMSE (Fig. 8 a) and correlation coefficients (Fig. 8 c). At Sodankylä however, the single-Alter shielded measurements were not significantly improved by the adjustments,

and the $PE_{0.1\ mm}$ values were actually slightly lower after adjustment. This indicates that at a field site such as Sodankylä, which was well sheltered from the wind in a forest clearing, such adjustments may not be necessary for single-Alter shielded gauges.

The single-Alter shielded results from the mountainous Haukeliseter, Formigal, and Weissfluhjoch sites demonstrated the same trend as the mountainous unshielded measurements, with larger errors in both the corrected and the uncorrected mountainous measurements, and much smaller RMSE and biases for the flat sites. The single-Alter shielded mean RMSE for the mountainous sites was only improved from 0.35 mm (42.8%) to 0.33 mm (41.6%) by adjustment, compared to the flat sites with a mean uncorrected RMSE of 0.18 mm (27.9%) that was improved to 0.13 mm (21.0%). For the single-Alter

shielded gauges, the mean of the absolute values of the biases for the mountainous sites were improved from 0.23 mm (29.0%) to 0.15 mm (18.4 %) and for other sites it was improved from 0.11 mm (18.0%) to 0.03 mm (4.7%). The general trend was that both before and after adjustment, the RMSE and the biases were much larger for the mountainous sites than for the other sites. The unadjusted mountainous measurements exhibited larger uncorrected errors, and these errors remained larger than the other, flatter sites after correction.

**3.3 Uncertainty of transfer functions**

In addition to the RMSE values shown in Fig. 5 a and 8 a, which describe the uncertainty of the adjusted measurements, the uncertainty of the $CE$ transfer functions were also estimated. As described by Fortin et al. (2008), the uncertainty of an adjusted precipitation measurement is affected by the uncertainty of the transfer function and the magnitude of both the precipitation measurement and the adjustment. $CE$ uncertainty estimates may be more difficult to interpret than the RMSE

included in Fig. 5 a and 8 a, but they can be used to calculate uncertainty estimates specific to new measurements and sites.

The RMSE values of the Eq. 3 transfer functions describing the unshielded $CE$ were 0.18 for both the gauge height and 10 m height wind speed transfer functions. The magnitude of the RMSE values for the different unshielded Eq. 3 and 4 transfer functions were similar and varied between 0.18 and 0.21. For the single-Alter shielded transfer functions, the uncertainty

varied from 0.18 to 0.19. When binned by wind speed, the resultant transfer function uncertainties were relatively insensitive to wind speed, and 0.2 can be used as a representative value for the uncertainty of all of the transfer functions, for both snow and mixed precipitation.

**3.4 High wind speed events**

As described in Section 2.2.8, for the testing of the transfer functions wind speeds greater than the wind speed threshold

were replaced with the wind speed threshold. Due to the inaccuracy of transfer functions at very high wind speeds, where data available to constrain the resultant fit were scarce, failure to implement the wind speed threshold could cause large errors due to over corrections at some sites. Although all measurements were used in the development of the transfer

functions, the high wind speed precipitation measurements were typically more accurately corrected using the wind speed threshold than the measured wind speed.

In addition, changing the wind speed threshold, and thereby changing the magnitude of the applied transfer function at high wind speeds, had a significant effect on the resultant site-specific errors and biases at some sites. Changing the gauge height wind speed threshold from 7.2 m s$^{-1}$ to 5 m s$^{-1}$, for example, improved the UN Weissfluhjoch bias from 35% to 16%, while simultaneously worsening the Haukeliseter bias from -7% to -23%, with similarly significant changes to the RMSE and other statistics at both sites. This suggests that local climatology may play an important role in determining the optimal individual wind speed threshold value.

Blowing snow may have also contributed to errors found at high wind speeds, with higher-than-normal catch efficiencies typically observed in blowing snow events (eg. Goodison, 1978; Schmidt, 1982). At Bratt's Lake, where the effects of blowing snow were quite pronounced and there was independent confirmation of blowing snow, these events were removed fairly easily by removing all of the high wind ($U_{10m} > 10$ m s$^{-1}$) events. At the other sites, blowing snow events were impossible to identify accurately. This is a potential problem for all windy sites, so all the high wind data were left in the data record to preserve a more realistic estimate of the uncertainties in the real-world, less-than-ideal conditions in which precipitation is typically measured.

**3.5 Gauge type**

Although the Pluvio$^2$ and the Geonor T-200B3 gauges differ in shape, with the Pluvio$^2$ having a larger and angled lip at the top of its inlet, no differences between the adjustments or catch efficiencies were observed between the gauge types. The dependence on siting and climate variability among the different sites was much larger than any potential effect of the gauge type. For the unshielded gauges, the mean bias of the adjusted Pluvio$^2$ measurements (from Weissfluhjoch, Formigal, and Sodankylä) was small (0.05 mm, or 6.4%), compared to the standard deviation of the mean site-biases from Pluvio$^2$ sites (0.21 mm, or 29.8%) and also the standard deviation of the mean site-biases from all sites (0.13 mm, or 17.5%). Likewise for the single-Alter gauges, the mean bias of the adjusted Pluvio$^2$ measurements from Weissfluhjoch, Formigal, and Sodankylä was small (0.03 mm, or 3.8%), compared to both the standard deviation of the Pluvio$^2$ biases (0.19 mm, or 26.6%) and the standard deviation of the biases from all the sites (0.11 mm, or 16.0%). The Geonor measurements from CARE, Haukeliseter, Caribou Creek, Marshall, and Bratt's Lake revealed similar results, with the variability in the biases by site much larger than the average bias of the Geonor sites. This is apparent in Fig. 5 and Fig. 8, noting that Pluvio$^2$ gauges were at Weissfluhjoch, Formigal, and Sodankylä and Geonor gauges were at CARE, Haukeliseter, Caribou Creek, Marshall, and Bratt's Lake. Likewise the RMSE, correlation coefficients, and $PE_{0.1mm}$ (Fig, 5 and Fig. 8) showed no significant correlation with gauge type, indicating that for these two weighing gauge types, errors due to siting, wind, and other causes were more significant than differences between the gauge types.

## 3.6 Transfer function verification using light precipitation events

The transfer functions were developed using 30-min events with $\geq 0.25$ mm of precipitation. However, the WMO-SPICE measurements also provided an opportunity to test the applicability of the transfer functions to light precipitation events, with the methods used to do this described in Section 2.2.5. Since Eq. 4 type transfer functions were not developed for liquid precipitation, only snow and mixed ($T_{air} < 2\ °C$) light precipitation events were included in this validation. Based on the Eq. 4 multi-site transfer function parameters obtained for gauge height wind speeds (Table 3), separate adjustment factors were applied to the light snow and mixed precipitation events. The adjusted snow and mixed precipitation data were merged together, and the test statistics were then calculated and examined by comparing the reference events to both the unadjusted and the adjusted light precipitation events.

The four sites considered in the assessment of light events were characterized by different climate conditions. 629 light precipitation events were observed at the low elevation, sub-arctic Sodankylä site. 361 light events were observed at CARE, with its continental climate. Haukeliseter, which is located in a mountainous region well above the tree line, experienced 260 light events. The smallest number of light events (62) was observed at the Bratt's Lake site, which is located in a prairie region with flat terrain.

Similar to the previous analysis, statistics were computed for each site by comparing the DFAR precipitation measurements to both the unadjusted and the adjusted light precipitation measurements (Fig. 9). The results for the unshielded and single-Alter shielded gauges were similar with regard to the benefit of the transfer function applications. The RMSE values were improved at 3 locations, with the only exception being the windiest site (Haukeliseter). The mean biases were improved, and often became positive, indicating that the applied transfer function corrected the underestimation in most cases. The percentage of events that agreed with the reference accumulation within the 0.05 mm range was also improved at all sites. After adjustment many more cases fell within the 0.05 mm error threshold, even for Haukeliseter, where 8% more SA and 24% more UN gauge observations were closer to the DFAR. The highest correlation between the reference and SA/UN gauges was observed at Sodankylä, with unadjusted correlations of 0.87/0.76 that were improved even further by adjustment. For Haukeliseter, the correlation decreased after adjustment due to overcorrected outliers. This was caused in part by events with unadjusted SA/UN measurements equal to or greater than the corresponding DFAR measurements. For example, following Eq. 4 the catch efficiency of a 7 m s$^{-1}$ wind speed event is 0.48, resulting in a multiplicative adjustment of 2.08, which would more than double the amount of precipitation and create a large overcorrection for such events. This effect is apparent in Fig. 10, which includes adjusted light precipitation event errors for all participating sites. At low wind speeds (< 3 m s$^{-1}$) the single-Alter shielded event errors (Fig 10 a) were closer to zero with less variation than the UN events (Fig 10 b), but at higher wind speeds some of the SA events were greatly overcorrected. These overcorrection errors were similar to the Weissfluhjoch SEDS errors shown in Fig. 6 a, although the scale of Fig. 10 a and Fig. 5 a y-axes differ.

### 3.7 Transfer function validation using 12 and 24 hour precipitation measurements

Because many historical precipitation measurements are only available for 12 and 24 h periods, the effects of applying the derived 30-min transfer functions to such measurement periods were examined. To help quantify the sensitivity of a transfer function to the accumulation time period used for its derivation, adjustments were also derived using both the 12 and 24 h periods. Error statistics for the adjusted 12 h measurements were estimated by applying both the 12 h transfer function and the 30 min transfer function to the same 12 h unshielded precipitation measurements (Fig. 11). The differences between the resultant RMSE were small, and varied from site to site, with the 30-min transfer function producing smaller RMSE on average (All, Fig. 11 a). The biases in the 30-min transfer functions were more negative than the 12 h transfer function biases, with the mean bias for all the measurements slightly underestimated in comparison to the near-zero 12 h transfer function bias (All, Fig. 11 b). Differences among the resultant correlation coefficients and $PE_{1.0\ mm}$ were relatively insignificant, and varied from site to site (Fig. 11 c and d). The unshielded 30-min transfer function was also applied to the 24 h precipitation periods, and was compared to a transfer function derived specifically for the 24 h precipitation measurements (Fig 12). Like the 12 h results, the transfer function developed on the 30-min precipitation measurements underestimated the 24 hour total precipitation when using 24 mean $T_{air}$ and wind speed, but the underestimate was relatively small, and was in some cases accompanied by improvements in the RMSE or $PE_{1.0\ mm}$. Also like the 12-h periods, site-to-site differences in the error statistics were much larger than differences between the 30-min and the 24 h adjustments. Similar analyses were performed on the single Alter measurements, and revealed similar results (data not shown).

The differences between the 30 min and the 12 or 24 h transfer function biases may have been caused by the fact that during precipitation it was either windier or colder than it was throughout the entire longer periods. The 30-min adjustments, for which the mean temperature and wind speed are more representative of conditions during precipitation, typically slightly under-corrected the longer precipitation periods, which may have experienced mean conditions that were typically warmer and less windy than the period when precipitation occurred within the 12 or 24 h period. This agrees with the analysis of high-frequency meteorological measurements from Jokioinen, Finland from the WMO Soild Precipitation Intercomparison (Goodison et al., 1998), which compared storm-average and 12 h average air temperature and wind speeds. Significant variability was noted, but the use of the average air temperature and wind speed rather than storm-average measurements for the application of transfer functions was demonstrated to slightly under-correct the precipitation measurements. This was because the storm-average temperature at Jokioinen was typically lower than the 12-h temperature, and the storm-average wind speed was typically higher than the 12-h wind speed.

Additional uncertainty was introduced into the 12 and 24 h precipitation measurements because it was not possible to screen for wind direction, as 12 and 24 h average wind directions were not always representative of the different wind directions recorded during precipitation. Because of this, from some locations these 12 and 24 h precipitation measurements may have

been affected by wind shadowing from neighbouring obstructions. In addition, the assumption of neutral conditions used to estimate either the gauge height or the 10 m height wind speed from the available wind speed measurements may not always have been valid for the 30-min wind speed measurements used to estimate the 12 and 24 h mean wind speeds. The assumption of neutral atmospheric conditions for the 30-min measurements is defensible because it is typically overcast

during precipitation, and clear skies are typically associated with strong surface heating and cooling and large vertical air temperature gradients. However longer time periods, such as 12 and 24 h, may include periods of precipitation and also periods of clear skies, and both stable and unstable surface layer conditions. However the effects of these different issues are likely small, especially given the good performance of the 30-min adjustments on longer time periods, but they may nevertheless merit closer inspection in future work.

**4 Conclusions**

Transfer functions developed and tested on eight separate sites were shown to reduce the biases in both unshielded and single-Alter shielded weighing gauge precipitation measurements. For the unshielded gauges, before adjustment the mean bias from all sites was -0.24 mm (-33.4%), and after adjustment it was 0.00 mm (1.1%). These unshielded and single-Alter adjustments are appropriate for use at other sites that require adjustment, and the uncertainties in the transfer functions have

been quantified using measurements from eight separate sites. The multi-site testing produced error estimates that were more useful, realistic, and representative than possible using only one or two sites. In addition, the methods used to create the precipitation data sets and derive these transfer functions can be used to help develop transfer functions at other sites. For example, this work can be used by other national weather services and researchers to develop new individual- and multi-site transfer functions for other gauges and shields of interest.

The mountainous sites of Formigal, Haukeliseter, and Weissfluhjoch were more difficult to correct, with more significant biases and RMSE remaining after adjustment. Higher wind speeds at the mountainous sites cannot fully explain this phenomenon, as only one of the mountainous sites was much windier than the other sites, and the mean site errors were not well-correlated with mean site wind speed. One possible explanation for this issue is that it may have been more difficult to

measure representative wind speeds at the mountainous sites. The Weissfluhjoch wind speed measurements provide some support for this hypothesis. Wind speed measurements at Weissfluhjoch were available from two different locations simultaneously during the winter of 2014-1015, and varied significantly from each other from all wind directions. Additional wind speed measurements, including more locations, more heights, and three-dimensional sonic anemometer measurements of the mean vertical wind speed might have helped identify the possible effects of large scale, standing circulations that

could have contributed to these discrepancies at the more complex sites. It is also possible that unique relationships between precipitation type and crystal habit and air temperature at the mountainous sites contributed to errors in the adjusted measurements. However, because one mountainous site was overcorrected (Weissfluhjoch), and the other two were

undercorrected (Formigal and Haukeliseter), it is not possible to recommend general modifications to the transfer functions for use in mountainous sites. These results indicate that the transfer functions developed and presented here should be evaluated at new testbeds in the mountains and complex terrain, and also in other areas subject to high winds and unusual precipitation.

As indicated by the RMSE values, significant differences between the DFAR measurements and both the unshielded and the single-Alter shielded measurements persisted after adjustment. For example, excluding gauges from the same three mountainous sites, the mean RMSE of the adjusted unshielded gauges was still 0.20 mm, or 31.5% of the mean 30-min precipitation. The mean RMSE of the adjusted single-Alter shielded gauge measurements at the flat sites (0.13 mm, or

21.0%) was lower than the unshielded gauge RMSE, confirming the increased accuracy provided by a single-Alter shield, but it was still significant. These errors in the adjusted measurements were presumably caused by a combination of factors, such as random spatial variability of precipitation across an individual site, sensor-induced noise in the precipitation measurements, the multiplicative effect of the transfer function corrections at high wind speeds (which can double or even triple both the amount of precipitation and the accompanying errors), and the effects of variability of crystal habit on catch

efficiency. This suggests that to produce more accurate measurements, a better understanding of the interaction of the snowflake trajectory past a given gauge/wind shield combination is needed. Recent Computational Fluid Dynamics (CFD) studies of airflow and snowflake trajectories past simple representations of gauges and Alter shields provide initial insights into this complex interaction (Colli et al., 2015; Theriault et al., 2012).

Two different types of weighing gauges – the Geonor T-200B3 and OTT Pluvio$^2$ – were included in the analysis. The two gauges differ in their principle of operation, with the Pluvio$^2$ using a load cell and the Geonor T-200B3 using vibrating wires. The shapes of the gauges are also different. However despite these differences, variations in siting and shielding were found to be more significant sources of uncertainty than the specific gauge type.

Overall, the adjusted single-Alter shielded measurements were found to be more accurate than the adjusted unshielded measurements. For all of the adjusted measurements, the average unshielded RMSE was 0.31 mm (42.8%), and for the single Alter shield the RMSE was 0.21 mm (30.6%). The errors in the unadjusted single-Alter measurements were also generally smaller than the unadjusted unshielded measurements. This is consistent with the design philosophy of shields, which is to reduce the horizontal wind speed inside the shield, and thereby reduce the effects of the gauge on the flow around

it.

The pre-SPICE transfer functions, which were created using both Marshall and Haukeliseter measurements for the single-Alter shielded gauge and only the Marshall site for the unshielded gauge (Kochendorfer et al., 2017), were quite similar to the more universal multi-site transfer functions developed here. This indicates that despite notable differences among the

eight different sites included in this study, robust transfer functions can be created using measurements from only one or two sites, provided that those sites are subject to typical catch efficiencies. For example, if Formigal was used to develop a transfer function for the Weissfluhjoch site (or vice versa) the resultant errors in the adjusted measurements would be large, as Formigal was on average undercorrected by the multi-site transfer function, and Weissfluhjoch was overcorrected. This also demonstrates the added value of using multiple sites to develop and test transfer functions.

The transfer functions also performed well on the light precipitation events, with improved biases and increases in the number of events that were within 0.05 mm of the DFAR. These results did not indicate that there was a significant change in the catch efficiency for light precipitation. They also confirmed the effectiveness of the transfer functions on these independent measurements, as the light precipitation events were excluded from the datasets used to create the transfer functions.

Application of the derived transfer functions to 12 and 24 h precipitation accumulations indicates that the transfer functions derived using 30 min periods can be applied to longer time periods. This is important for historic precipitation records, which are often only available every 12 or 24 h. In general, the sensitivity to the period chosen to derive the transfer function was small, and it varied from site to site. Most importantly, when tested on 12 and 24 hr precipitation measurements, the differences between the error statistics describing transfer functions derived from 30 min, 12 h, and 24 h accumulations was in all cases smaller than the variability between sites. This indicates that when these transfer functions are applied to new sites, errors due to the variability in climatology will be larger than errors caused by longer measurement frequencies.

**Acknowledgements**

We thank Hagop Mouradian of Environment and Climate Change Canada for contributing the mapped site locations (Fig. 1). This research was funded by the Korean Ministry of Land, Infrastructure and Transport through a grant (16AWMP-B079625-03) from the Water Management Research Program. We also thank Eckhard Lanzinger, Vincent Fortin, and Kay Helfricht for providing thoughtful reviews of the originally-submitted version of this manuscript, and greatly improving the quality of this paper.

**Disclaimers**

Many of the results presented in this work were obtained as part of the Solid Precipitation Intercomparison Experiment (SPICE) conducted on behalf of the World Meteorological Organization (WMO) Commission for Instruments and Methods of Observation (CIMO). The analysis and views described herein are those of the authors at this time, and do not necessarily represent the official outcome of WMO-SPICE. Mention of commercial companies or products is solely for the purposes of

information and assessment within the scope of the present work, and does not constitute a commercial endorsement of any instrument or instrument manufacturer by the authors or the WMO.

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

**Figures**

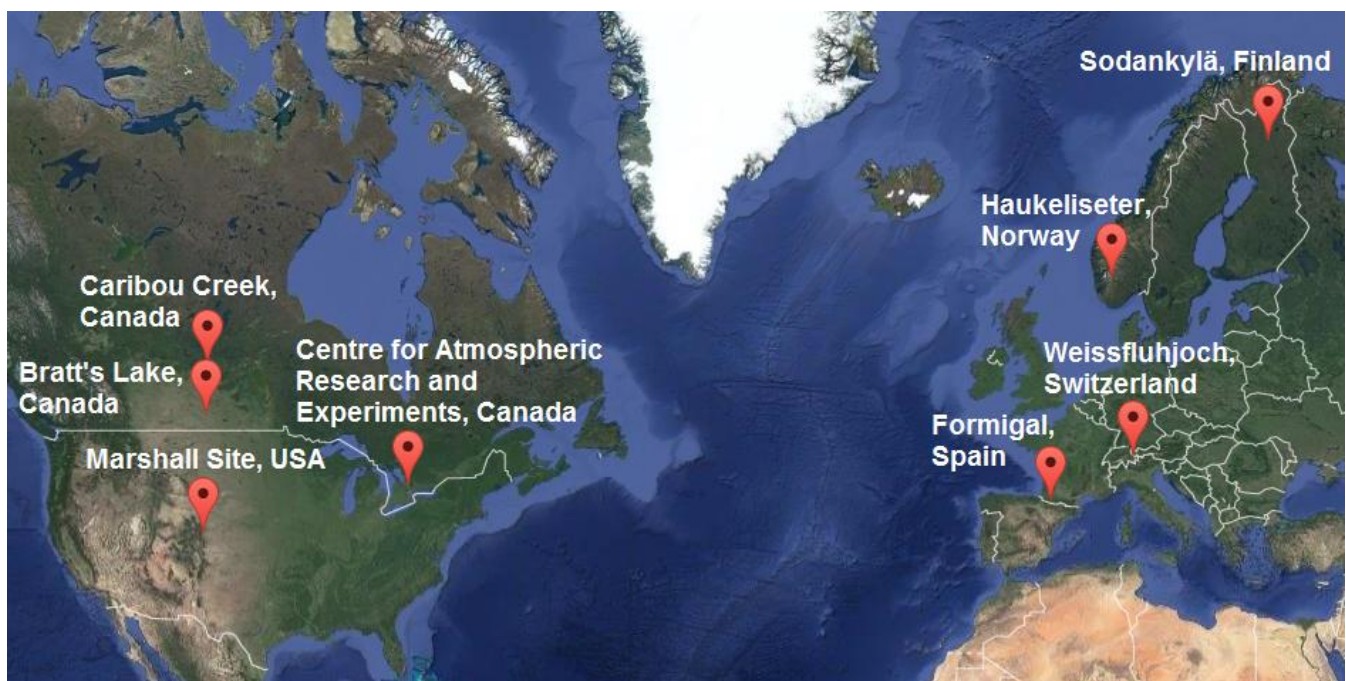

**Figure 1. SPICE testbeds included in this study.**

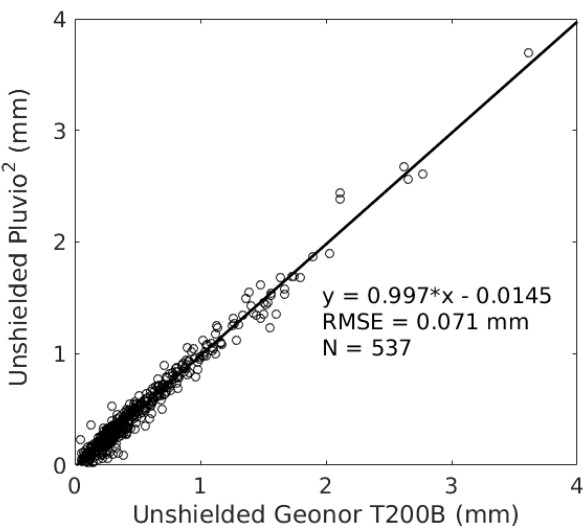

5    **Figure 2. Comparison of unshielded Pluvio² and Geonor gauges at the CARE testbed during WMO-SPICE.**

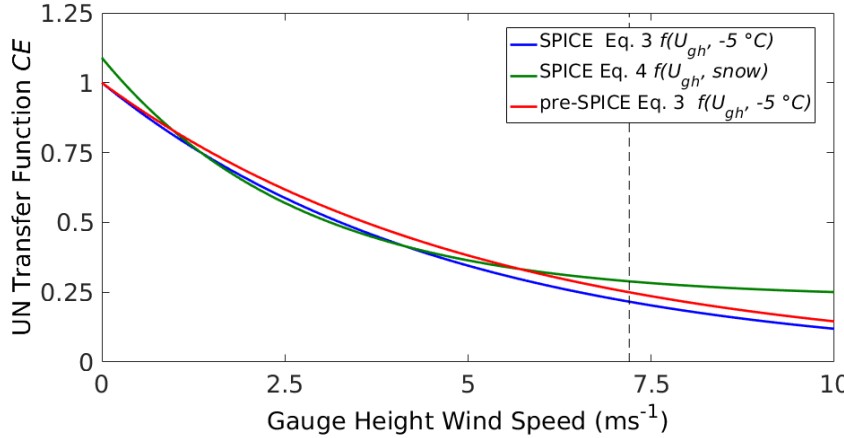

**Figure 3.** Transfer functions describing the unshielded (UN) catch efficiency (*CE*) as a function of the gauge height wind speed ($U_{gh}$). The Eq. 3 results were produced by modelling *CE* with respect to wind speed at $T_{air}$ = -5 °C, and both the Kochendorfer et al. (2017) pre-SPICE (red line) and the current results (blue line) are shown. The Eq. 4 snow ($T_{air}$ < -2 °C) results (green line) are also shown. The vertical dashed line indicates the wind speed threshold (7.2 m s$^{-1}$) above which the transfer function should be applied by forcing $U_{gh}$ to 7.2 m s$^{-1}$.

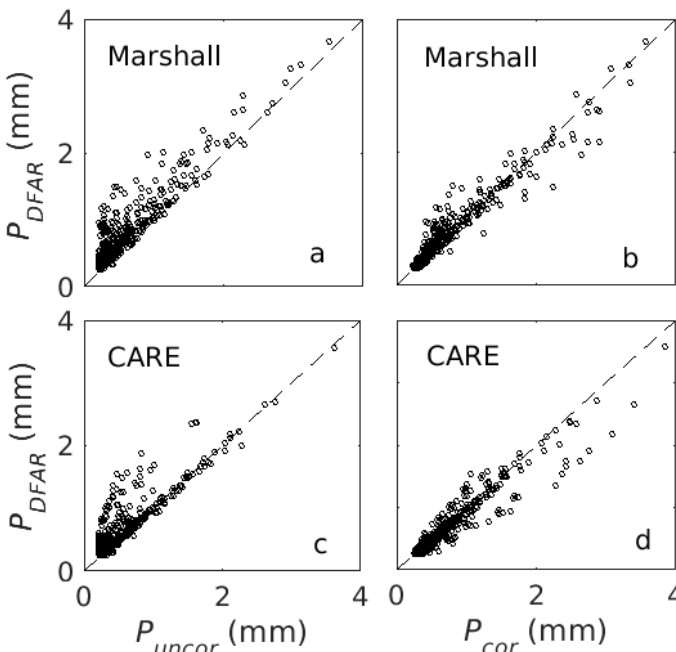

**Figure 4.** Comparison of uncorrected, unshielded precipitation measurements ($P_{uncor}$, a, c) and corrected unshielded precipitation measurements ($P_{cor}$, b, d) with DFAR precipitation measurements ($P_{DFAR}$) at the Marshall (a, b) and CARE (c, d) testbeds. The dashed line describes a 1:1 relationship.

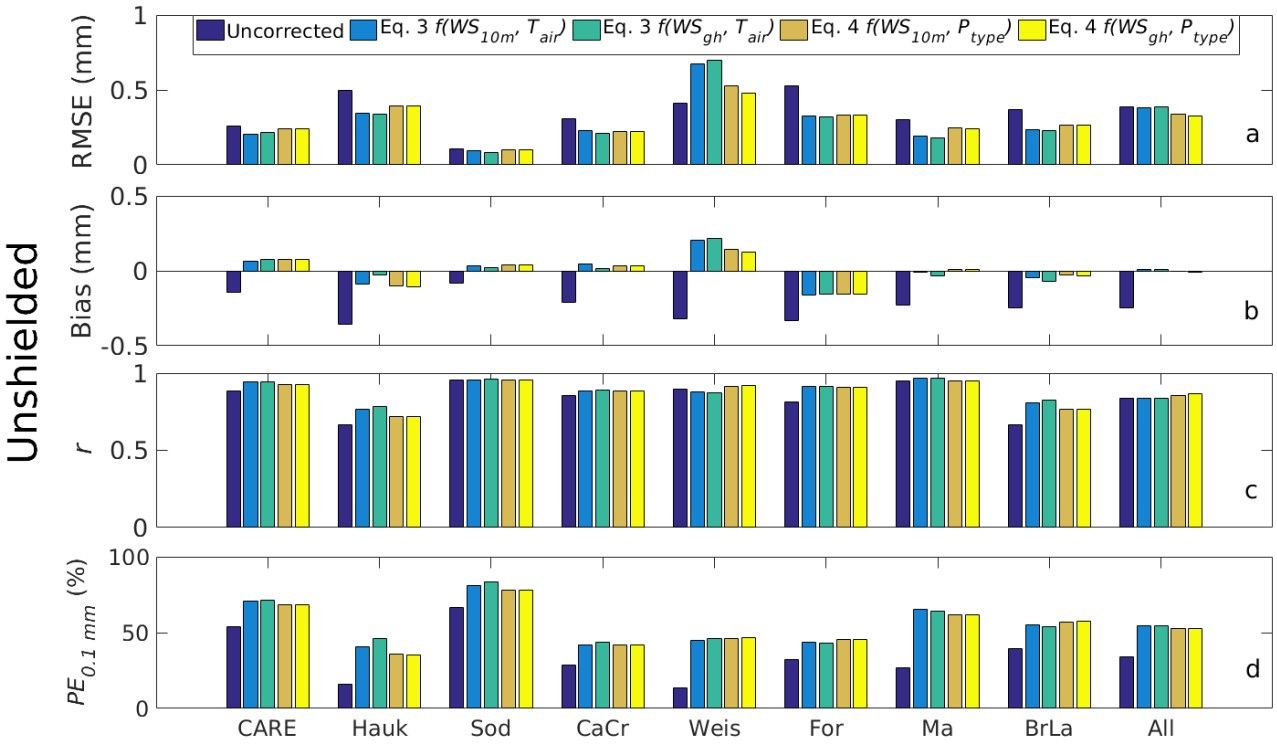

**Figure 5. Unshielded gauge statistics for all 8 sites, calculated from the difference between the DFAR precipitation and both the uncorrected (dark blue) and the corrected precipitation. The corrected precipitation measurements were based on the Eq. 3 transfer function for both the 10m wind speed ($U_{10m}$, light blue) and the gauge height wind speed ($U_{gh}$, green), and the Eq. 4 transfer function for both the 10m wind speed (tan) and the gauge height wind speed (yellow). The RMSE (a), mean bias (b), correlation coefficient ($r$, Panel c), and the percent of events that were within 0.1 mm ($PE_{0.1\ mm}$, Panel d) of the reference are shown for individual sites and all of the measurements combined (All).**

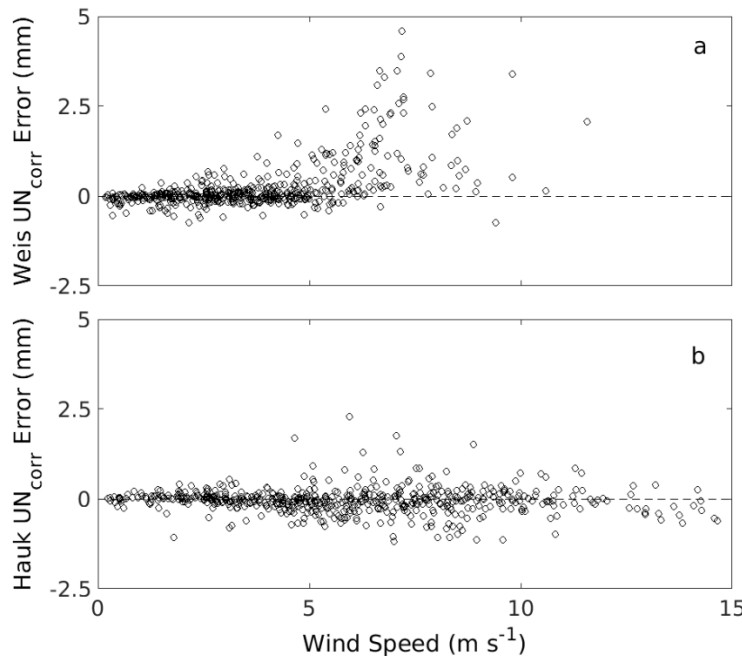

Figure 6. Errors in the 30-min precipitation, estimated from the difference between the DFAR and the corrected, unshielded gauges at Weissfluhjoch (a) and Haukeliseter (b). The dashed line describes where the error is equal to zero.

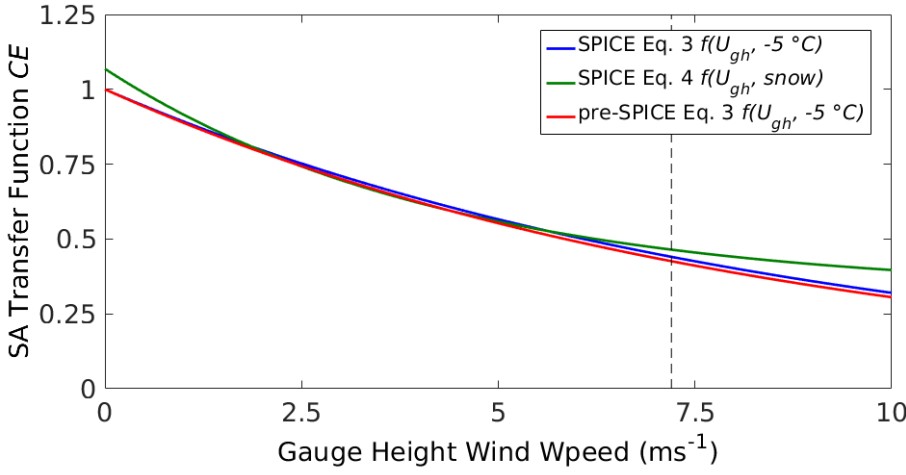

5    Figure 7. Transfer functions describing the single Alter (SA) catch efficiency ($CE$) as a function of the gauge height wind speed ($U_{gh}$). The Eq. 3 results were produced by modelling $CE$ with respect to wind speed at $T_{air}$ = -5 °C, and both the Kochendorfer et al. (2017) pre-SPICE (red line) and the current results (blue line) are shown. The Eq. 4 snow ($T_{air}$ < -2 °C) results (green line) are also shown. The vertical dashed line indicates the wind speed threshold (7.2 m s$^{-1}$) above which the transfer function should be applied by forcing $U_{gh}$ to 7.2 m s$^{-1}$.

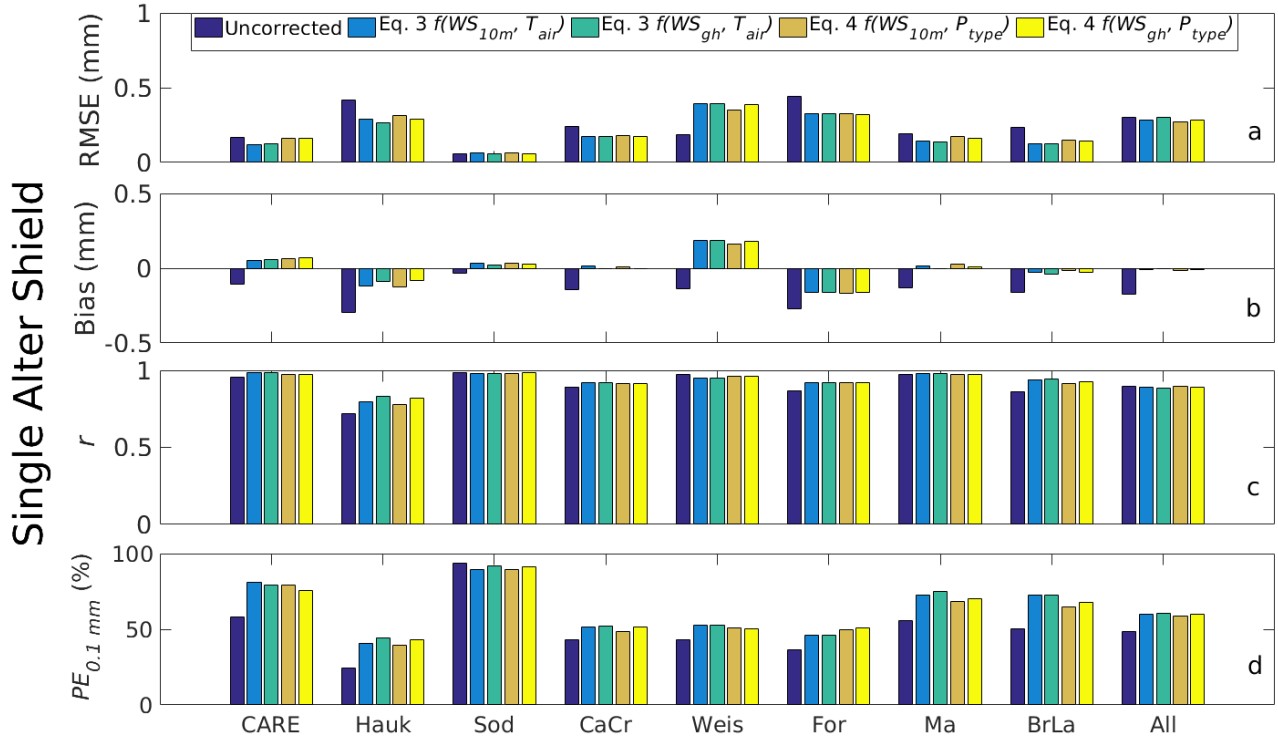

**Figure 8. Single Alter gauge statistics for all 8 sites, calculated from the difference between the DFAR precipitation and both the uncorrected (dark blue) and the adjusted precipitation. The adjusted precipitation measurements were based on the Eq. 3 transfer function for both the 10m wind speed ($U_{10m}$, light blue) and the gauge height wind speed ($U_{gh}$, green), and the Eq. 4 transfer function for both the 10m wind speed (tan) and the gauge height wind speed (yellow). The RMSE (a), mean bias (b), correlation coefficient ($r$, Panel c), and the percent of events that were within 0.1 mm ($PE_{0.1\ mm}$, Panel d) of the reference are shown for individual sites and all of the measurements combined (All).**

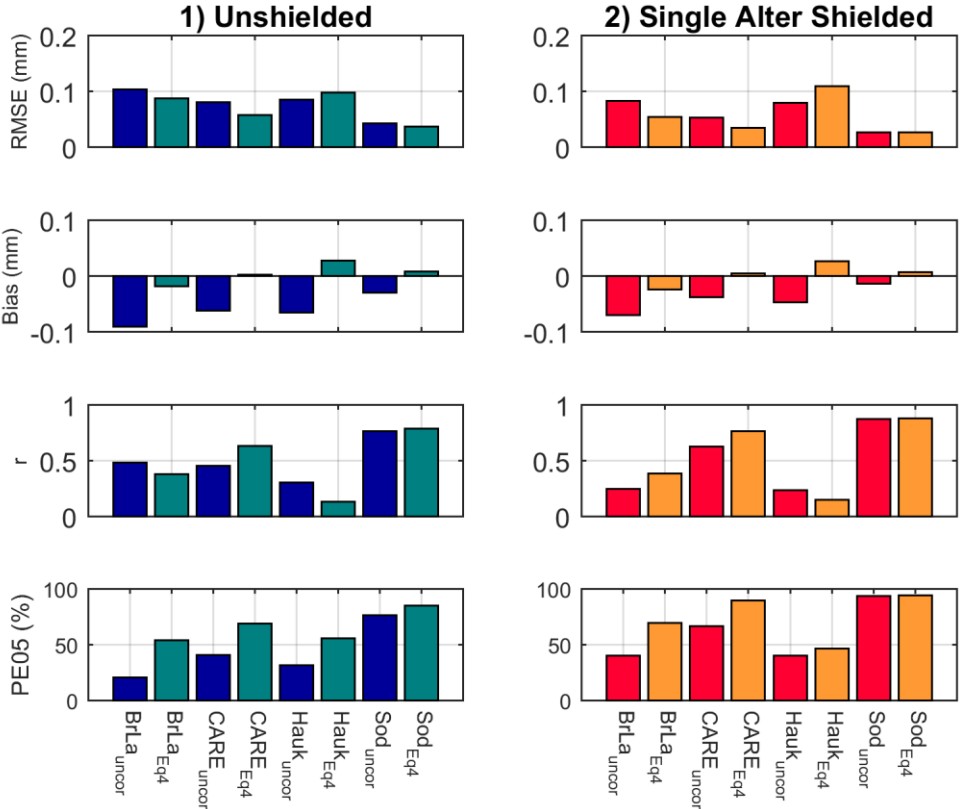

**Figure 9. SLEDS gauge statistics for 4 sites, calculated from the difference between the DFAR precipitation and both the uncorrected (dark blue for (1) Unshielded / red for (2) Single Alter Shielded) and the adjusted (light blue for (1) Unshielded / orange for (2) Single Alter Shielded) precipitation using Eq. 4. The RMSE, mean bias, correlation coefficient (r), and the percent of events within 0.05 mm range (PE05 mm) of the reference are shown for the individual sites.**

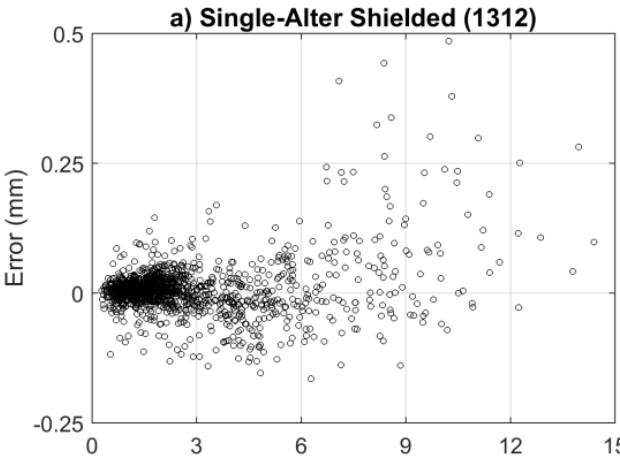

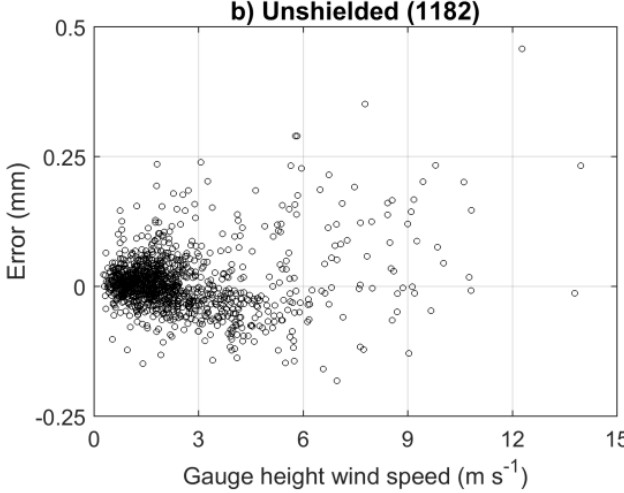

**Figure 10. Errors in the 30-min SLEDS precipitation, estimated from the difference between the DFAR and the corrected single-Alter shielded (a) and unshielded (b) gauges at the CARE, Bratt's Lake, Haukeliseter and Sodankylä sites.**

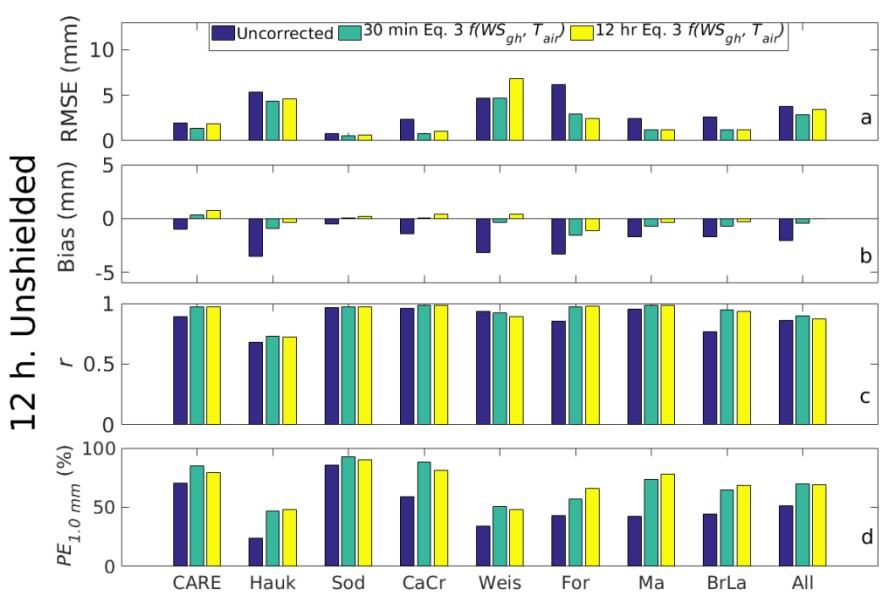

**Figure 11. Error statistics for 12 h unshielded precipitation measurements that are uncorrected (blue), corrected using the 30-min derived transfer functions (green), and corrected using the 12 h derived transfer functions (yellow) are compared.**

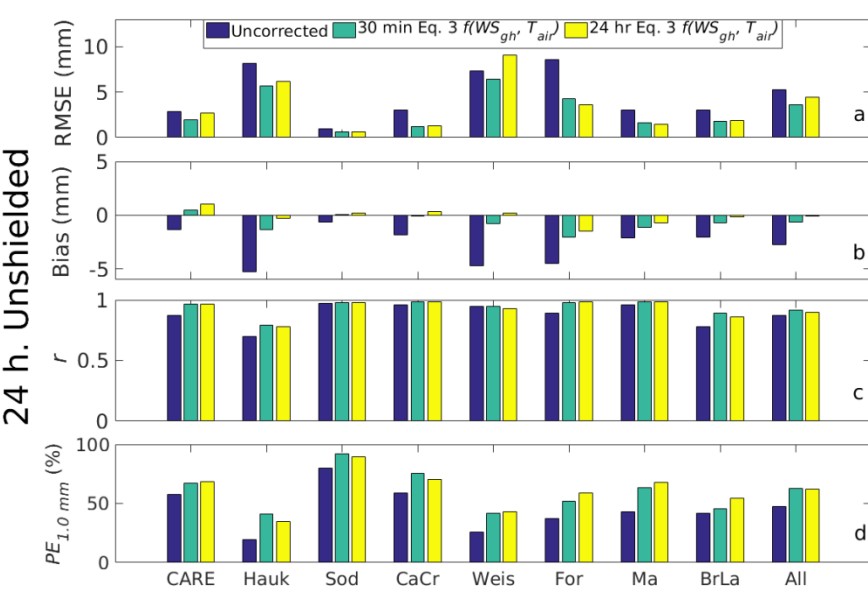

5    **Figure 12. Error statistics for 24 h unshielded precipitation measurements that are uncorrected (blue), corrected using the 30-min derived transfer functions (green), and corrected using the 12 h derived transfer functions (yellow) are compared.**

**Tables**

| Site | Country | Abbr | Elev. | Lat. | Mean $U_{gh}$ | Max $U_{gh}$ | Mean $T_{air}$ | $N_{UN}$ | $N_{SA}$ |
|------|---------|------|-------|------|-----------|-----------|------------|-------|-------|
| CARE | Canada | CARE | 251 m | 44.23° | 3.2 m s$^{-1}$ | 8.2 m s$^{-1}$ | -3.3 °C | 484 | 364 |
| Haukeliseter | Norway | Hauk | 991 m | 59.81° | 6.7 m s$^{-1}$ | 20.6 m s$^{-1}$ | -1.7 °C | 565 | 635 |
| Sodankylä | Finland | Sod | 179 m | 67.37° | 1.6 m s$^{-1}$ | 4.0 m s$^{-1}$ | -2.1 °C | 507 | 507 |
| Caribou Creek | Canada | CaCr | 519 m | 53.94° | 2.6 m s$^{-1}$ | 7.2 m s$^{-1}$ | -6.3 °C | 413 | 388 |
| Weissfluhjoch | Switzerland | Weis | 2537 m | 46.83° | 3.8 m s$^{-1}$ | 11.6 m s$^{-1}$ | -7.2 °C | 508 | 537 |
| Formigal | Spain | For | 1800 m | 42.76° | 2.3 m s$^{-1}$ | 6.0 m s$^{-1}$ | -0.7 °C | 669 | 656 |
| Marshall | USA | Ma | 1742 m | 39.59° | 2.8 m s$^{-1}$ | 10.2 m s$^{-1}$ | -2.0 °C | 466 | 459 |
| Bratt's Lake | Canada | BrLa | 585 m | 50.20° | 4.4 m s$^{-1}$ | 7.3 m s$^{-1}$ | -1.5 °C | 168 | 182 |

Table 1. Site names, abbreviations (Abbr), latitude, mean gauge height wind speed ($U_{gh}$), maximum $U_{gh}$, mean air temperature ($T_{air}$), and the number of 30-min unshielded ($N_{UN}$) and single-Alter shielded ($N_{SA}$) precipitation measurements within the 2013-2015 winter seasons. Statistics were calculated only from periods of precipitation included in the analysis, and therefore differ from actual site climatologies.

| Shield | $f(U_{gh}, T_{air})$ | | | $f(U_{10m}, T_{air})$ | | | $U_{thresh}$ | | $n$ |
|--------|------|------|------|------|------|------|--------|--------|-----|
| | $a$ | $b$ | $c$ | $a$ | $b$ | $c$ | $U_{10m}$ | $U_{gh}$ | |
| UN | 0.0785 | 0.729 | 0.407 | 0.0623 | 0.776 | 0.431 | 9 m s$^{-1}$ | 7.2 m s$^{-1}$ | 3774 |
| SA | 0.0348 | 1.366 | 0.779 | 0.0281 | 1.628 | 0.837 | 9 m s$^{-1}$ | 7.2 m s$^{-1}$ | 3725 |

Table 2. $f(U, T_{air})$ transfer function (Eq. 3) coefficients for gauge-height (gh) and 10 m wind speeds. Also shown are the wind speed thresholds for which the transfer functions are valid ($U_{thresh}$), and the number of 30-min periods used to create the transfer functions ($n$).

| Shield | Phase | $f(U_{gh})$ | | | $f(U_{10m})$ | | | $U_{thresh}$ | | $n$ |
|--------|-------|------|------|------|------|------|------|--------|--------|-----|
| | | $a$ | $b$ | $c$ | $a$ | $b$ | $c$ | $U_{10m}$ | $U_{gh}$ | |
| UN | mixed | 0.641 | 0.236 | 0.356 | 0.624 | 0.185 | 0.364 | 9 m s$^{-1}$ | 7.2 m s$^{-1}$ | 1245[15] |
| UN | solid | 0.860 | 0.371 | 0.229 | 0.865 | 0.298 | 0.225 | 9 m s$^{-1}$ | 7.2 m s$^{-1}$ | 1968 |
| SA | mixed | 0.668 | 0.132 | 0.339 | 0.821 | 0.077 | 0.175 | 9 m s$^{-1}$ | 7.2 m s$^{-1}$ | 1297 |
| SA | solid | 0.728 | 0.230 | 0.336 | 0.742 | 0.181 | 0.322 | 9 m s$^{-1}$ | 7.2 m s$^{-1}$ | 1929 |

Table 3. $f(U)$ transfer function (Eq. 4 ) coefficients for mixed and solid precipitation, developed for both gauge-height (gh) and 10 m wind speeds. Also shown are the wind speed thresholds for which the transfer functions are valid ($U_{thresh}$), and the number of 30-min periods used to create the transfer functions ($n$).