# Peer review of "Analysis of single-Alter shielded and unshielded measurements of mixed and solid precipitation from WMO-SPICE"

_Hydrology and Earth System Sciences, 2016_

## Referee Comment (RC1) · V. Fortin (Referee) · 24 Jan 2017

This is a much needed paper that addresses a very important problem in meteorology, climatology and hydrology: dealing with gauge undercatch from all-weather gauges in operations around the world. The paper is well written and nicely builds upond the published literature on the subject. It makes use of a unique dataset carefully gathered during the SPICE project. I strongly recommend its publication. I however do have a few comments that I would like to see addressed before the paper is published in HESS.

My most important comment is that the results may not be readily applicable, because of the authors' decision to derive transfer functions that require 30 minute data. Subhourly data is not easily accessed in real-time, and can be very difficult if not impossible to obtain for archived data. Even hourly data is hard to obtain. And when it is accessible, it is often not quality controlled at this frequency. The authors are strongly encouraged to discuss how their method could be applied to data that is only available at lower frequencies (hourly, three-hourly, six-hourly, twelve-hourly and daily).

The authors acknowledge that significant uncertainty remains after bias correction on the precipitation amount, even if the method does a reasonable job of controlling the bias. The authors should ideally communicate this uncertainty by publishing, together with the transfer function, an estimate of the standard error of the catch efficiency, as was done for example by Fortin et al. (2008), Hydrol. Proc.

Using the method proposed in this paper by the authors, I hope that bias-corrected precipitation data can soon be used in an optimal manner by land-surface data assimilation systems in cold regions. Such systems are routinely used to initialize land-surface, meteorological and hydrological forecasting systems. However, in a data assimilation system it is crucial to accurately estimate the standard error of the observations. Information on the standard error of the catch efficiency is obviously crucial for this purpose. This is why I strongly recommend that the authors propose an estimate for the standard error of the catch efficiency together with the transfer functions.

**Minor comments:**

Equation (2) The equation is incorrect. Wind speed is proportional, not approximately equal to  $log[(z - d)/z_0]$ . It should be mentioned that this equation assumes neutral stability conditions.

Section 2.2.6 The authors need to better justify lumping together data from the Pluvio and Geonor gauges.

---

## Referee Comment (RC3) · E. Lanzinger (Referee) · 27 Apr 2017

General Comments:

Large measuring uncertainties for solid precipitation measurements are a major issue in hydrology. The paper "Errors and adjustments for single-Alter shielded and unshielded weighing gauge precipitation measurements from WMO-SPICE" by John Kochendorfer et al. is clearly a substantial contribution to improved solid precipitation measurements.

Besides improved, robust correction formulae there is also an estimate of the remaining uncertaitiy given. Outliers in the results are clearly explained and hints for improvement

are given. The text is well structured and language, explanations, graphs and tables are very clear and don't need any change. I have only minor questions in some parts That I address below.

The intention of this paper is to provide a simple and practical method for correcting solid precipitation data by ancillary data that is available at most of the meteorological stations world wide. The scientific significance, quality, and the presentation quality are therefore excellent.

Specific Comments:

P. 1, lines 23/24: Choose a more appropriate word for "differences". What you mean here are "errors" or "measuring uncertainties". The word "difference" is too neutral, i.e. different amounts of precipitaion could indeed occur in different regions without being an error.

P. 2, line 4: In "...changes in the velocity of the air around the gauge..." velocity and direction of the airflow could be added, as the air flow is bended around the gauge which is also leading to an uplift of light particles over the gauge orifice.

P. 2, line 19/20: Could you eventually find the original citations that served as a rationale for the WMO decision of 2010? Your citation is of 2012. you could use

P. 2, lines 27-29: It is not understandable for everybody, what the term "WMO-SPICE weighing precipitation gauges" means. I suggest two sentences: The focus of the work described below is on unshielded and single-Alter-shielded weighing precipita- tiongauges. Based on results of a previous CIMO survey (Nitu and Wong, 2010), WMO-SPICE selcted two weighing gauges which represent the two most ubiquitous configurations used in national networks for the measurement of solid precipitation.

P. 2, line 32: DFIR needs a citation.

P. 3, lines 7-8: "Some of this variability is driven by differences in ice crystal shape (habit), mean hydrometeor fall velocity, and hydrometeor size...". Actually hydrometeor

fall velocity is a function of its size, shape and mass (or mass density). Therefore it should not be mentioned side by side with the other variables. Hydrometeor shape and density are very difficult to measure for each particle, and are therefore generally not available. But some disdrometers provide hydrometeor size and fall velocity. By using fall velocity you implicitely take into account crystal shape and mass density.

Please check again the given citation (Thériault et al. 2012) for details.

P. 9, line 3: "mixed precipitation was defined as..." I wonder whether "mixed precipitation" is a good term here, because you would expect a mixture of rain an snow for each of the events. I guess most of the events were snow only? Isn't the temperature used to distinguish between wet and dry snow? I would suggest "wet snow/mixed precipitation" and for the colder regime eventually "dry snow."

P.9, line 5: I suggest to delete the phrase "and the negligible magnitude of the liquid precipitation adjustment", because for drizzle the CE in windy conditions is negatively affected and needs correction. What you might have in mind is that the corrections for snow are generally larger. But as the fall speed of small droplets and small snow particles are very close it is clear that both will be affected by wind in a comparable way.

P. 13, line 1: PE was improved by the application of tranfer functions for all sites, except one: Sodankylä. It is really a very very small difference, but it could be mentioned that in a site that is well shielded by trees and where there are generally low wind speeds, a SA shield is already sufficient and no further correction is needed. As you mentioned, the correction curves are not precise at the low and high wind speed end. Could this be a reason, why the Sondakylä results with SA shield are getting a bit worse after correction?

With the same argument, that you require to keep the correction constant for wind speeds higher than 7.2 m/s you could think about keeping the correction constant to 1 (or some value close to 1) for wind speeds lower than a threshold. By these two means

you get closer to the sigmoid curves published elsewhere.

Typos and linguistic comments:

P. 5 line 11: rationale

P. 5, line 15/16: "false precipitation error" sounds a bit like "false error". I suggest "false accumulation" instead.

P. 6, line 2: "10 m wind speed". I suggest "wind speed at 10 m height"

P. 8, line 4: threshold of 0.1 mm in 30 min was chosen. . . the 30 min could be added.

P. 17, line 1: ...reduce the horizontal wind speed impacting...

---

## Author Comment (AC2) · 22 May 2017

**Response to V. Fortin.**

This is a much needed paper that addresses a very important problem in meteorology, climatology and hydrology: dealing with gauge undercatch from all-weather gauges in operations around the world. The paper is well written and nicely builds
5   upon the published literature on the subject. It makes use of a unique dataset carefully gathered during the SPICE project. I strongly recommend its publication. I however do have a few comments that I would like to see addressed before the paper is published in HESS.

My most important comment is that the results may not be readily applicable, because of the authors' decision to derive
10   transfer functions that require 30 minute data. Sub-hourly data is not easily accessed in real-time, and can be very difficult if not impossible to obtain for archived data. Even hourly data is hard to obtain. And when it is accessible, it is often not quality controlled at this frequency. The authors are strongly encouraged to discuss how their method could be applied to data that is only available at lower frequencies (hourly, three-hourly, six-hourly, twelve-hourly and daily).

**Authors' response:** Fortin raises an important issue here regarding the relationship between the time period over which a
15   precipitation measurement is recorded and the resultant catch efficiencies. This topic was the subject of much discussion and analysis among the authors of this manuscript and the other participants in WMO-SPICE, but this work was admittedly not originally reflected in the present manuscript. 30-min data were used to derive the transfer functions because this period is short enough to allow for representative wind speed and air temperature measurements while simultaneously being long enough to allow for significant and measurable solid precipitation to accumulate, but this does not mean that the transfer
20   functions can only be applied to 30-min measurements. To address this, additional analysis has been performed on 12 and 24 h precipitation measurements. 12 and 24 h precipitation accumulations were created, and the 30-min transfer functions were applied to them. For the sake of comparison, transfer functions specific to the 12 and 24 h accumulations were also created and applied to the appropriate measurements. This analysis is described in the new Methods Section 2.2.6, an additional paragraph in Methods Section 2.2.9, the new Results Section 3.7, and an additional paragraph in the Conclusions. These
25   changes are documented in the track-changes version of the revised manuscript, which has been provided along with our responses to the reviews. Four new figures have also been created describing the 12 and 24 h precipitation measurement errors (Figures A1 − A4), demonstrating that the transfer functions derived from the 30 min measurements are appropriate for these longer time periods.  However, for the sake of brevity only the figures describing the unshielded measurements (Figures A1 and A3) are included in the revised manuscript.

30

The authors acknowledge that significant uncertainty remains after bias correction on the precipitation amount, even if the method does a reasonable job of controlling the bias. The authors should ideally communicate this uncertainty by publishing,

together with the transfer function, an estimate of the standard error of the catch efficiency, as was done for example by Fortin et al. (2008), Hydrol. Proc.

**Authors' response:** This is indeed a good point. The uncertainty has now been calculated for all of the transfer functions (Table A1 below) and a summary of the results has been added to the manuscript (Section 3.3). These results show that the uncertainty is approximately 0.2 for all of the functions tested. Other testing performed in a separate manuscript submitted to HESS in April 2017 (which has not yet been published for discussion) show that the uncertainty of the transfer function is also fairly insensitive to the wind speed.

Using the method proposed in this paper by the authors, I hope that bias-corrected precipitation data can soon be used in an optimal manner by land-surface data assimilation systems in cold regions. Such systems are routinely used to initialize land-surface, meteorological and hydrological forecasting systems. However, in a data assimilation system it is crucial to accurately estimate the standard error of the observations. Information on the standard error of the catch efficiency is obviously crucial for this purpose. This is why I strongly recommend that the authors propose an estimate for the standard error of the catch efficiency together with the transfer functions.

**Authors' response:** We agree. See the response to the comment above.

Minor comments: Equation (2) The equation is incorrect. Wind speed is proportional, not approximately equal to $\log[(z - d)/z0]$. It should be mentioned that this equation assumes neutral stability conditions.

**Authors' response:** Thank you! The $\approx$ symbol has been replaced with a proportional symbol, and the assumption of neutral stability has been noted in the revised manuscript.

Section 2.2.6 The authors need to better justify lumping together data from the Pluvio and Geonor gauges

**Authors' response:** Fortin brings up an important point here. Section 2.1 (pg. 4, ln. 9 -13) does include some justification for combining the Pluvio and Geonor gauge data, but further justification has been added to this section of the revised manuscript. The CARE site had both unshielded and single-Alter shielded Pluvio[2] and Geonor gauges. In response to this comment, these paired measurements were compared more closely. Fig A5 shows the results of the comparison between the unshielded gauge measurements. 389 single-Alter shielded Pluvio[2] and Geonor measurements were also compared, resulting in a slope of 1.01, offset of -0.002, and a RMSE of 0.085 mm (not shown); fewer single-Alter shielded measurements were available for comparison with each other due to the proximity of the DFIR to the single-Alter shielded Geonor gauge at the CARE testbed. In addition, Eq. 4 type solid precipitation transfer functions were created independently for each of the two types of single-Alter shielded gauges and also for the two unshielded gauges at CARE (e.g. Fig. A6), and no significant differences were found between the wind-speed responses of the different gauge types. Fig. A5 and a summary of these comparisons have been added to the first Methods Section of the revised manuscript. Fig. A6 required some additional

analysis to create (the version included here is in fact somewhat preliminary), requiring accompanying explanation and methods, and is arguably not important enough or central enough to the main point of the manuscript to merit its addition.

**Figures**

[Figure]

5      **Figure A1. Error statistics for 12 h unshielded precipitation measurements that are uncorrected (blue), corrected using the 30-min derived transfer functions (green), and corrected using the 12 h derived transfer functions (yellow) are compared.**

[Figure]

**Figure A2. Error statistics for 12 h single-Alter shielded precipitation measurements that are uncorrected (blue), corrected using the 30-min derived transfer functions (green), and corrected using the 12 h derived transfer functions (yellow) are compared.**

[Figure]

5    **Figure A3. Error statistics for 24 h unshielded precipitation measurements that are uncorrected (blue), corrected using the 30-min derived transfer functions (green), and corrected using the 12 h derived transfer functions (yellow) are compared.**

[Figure]

**Figure A4.** Error statistics for 24 h single-Alter shielded precipitation measurements that are uncorrected (blue), corrected using the 30-min derived transfer functions (green), and corrected using the 12 h derived transfer functions (yellow) are compared.

[Figure]

**Figure A5.** Comparison of unshielded Pluvio[2] and Geonor gauges at the CARE testbed during WMO-SPICE.

[Figure]

**Figure A6. Catch efficiency (*CE*) measurements from unshielded Pluvio[2] (black circles) and Geonor (red circles) solid precipitation ($T_{air}$ < -2 °C) measurements at the CARE testbed during WMO-SPICE. Eq. 4 has been fit to each measurement type (solid lines). Errors were estimated using the RMSE of the *CE* transfer functions, which were 0.13 for the Pluvio[2] and 0.12 for the Geonor.**

**Tables**

| Configuration | Transfer function | Wind speed | *CE* RMSE |
|---|---|---|---|
| Unshielded | Eq. 3 | Gauge height | 0.18 |
| Unshielded | Eq. 4, mixed | Gauge height | 0.20 |
| Unshielded | Eq. 4, snow | Gauge height | 0.19 |
| Unshielded | Eq. 3 | 10 m | 0.18 |
| Unshielded | Eq. 4, mixed | 10 m | 0.21 |
| Unshielded | Eq. 4, snow | 10 m | 0.19 |
| Single Alter | Eq. 3 | Gauge height | 0.18 |
| Single Alter | Eq. 4, mixed | Gauge height | 0.19 |
| Single Alter | Eq. 4, snow | Gauge height | 0.19 |
| Single Alter | Eq. 3 | 10 m | 0.18 |
| Single Alter | Eq. 4, mixed | 10 m | 0.19 |
| Single Alter | Eq. 4, snow | 10 m | 0.19 |

**Table A1. Transfer function uncertainty, expressed as the RMSE of the function used to describe catch efficiency (*CE*).**

---

## Author Comment (AC3) · 22 May 2017

**Response to K. Helfricht**

General comments

The paper "Errors and adjustments for single-Alter shielded and unshielded weighing gauge precipitation measurements from WMO-spice" by John Kochendorfer et al. is published as a part of the HESS special issue presenting results of the recent WMO initiative evaluating the catch efficiency from different gauge types. The paper contributes to the present efforts of adjustments of precipitation undercatch for a wide range of applications in climatology and hydrology as well as real-time corrections for nowcast and short-term forecast applications. On the basis of well measured data from eight locations including lowland and mountain stations it presents transfer functions which can be used to adjust 30 minute precipitation gauge data for undercatch in scientific studies and in operational services. The paper is well structured and concisely written. It presents the literature on this subject comprehensively. The paper is worth publishing in HESS with a few minor corrections.

**Authors' response:** Thank you.

Specific comments

(1) The authors used aggregated 30 minute precipitation data to develop the transfer functions. However, the authors should discuss if the presented transfer functions are also valid adjusting precipitation data of higher or lower time intervals, e.g. 10 minute, hourly or daily. Deviations can be expected caused by different mean wind speeds. This may be achieved by calculating adjusted precipitation for the sub-daily time intervals and comparing the daily aggregated values.

**Authors' response:** Deviations caused by different mean wind speeds are indeed possible for longer time periods. The validity of the transfer functions derived from 30-min measurements when applied to longer time periods (12 h and 24 h) has been addressed in the revised manuscript and the response to Fortin. The approach used was to create 12 and 24 h precipitation datasets, apply the 30-min derived transfer functions, and also derive and apply 12 and 24 h derived transfer functions. Because the reference amount of precipitation is known and has been measured in these different time intervals, it is not necessary to take the suggested step of summing up from shorter intervals to longer intervals in order to evaluate the effects of varying time interval, although this is a good suggestion. Accumulation periods shorter than 30 min have not been evaluated. One of the challenges in evaluating different time periods is that datasets derived for different time periods following the WMO-SPICE event selection criteria were independent of each other. For example the sum of all of the 30-min event accumulations in a day will not necessarily be equal to the corresponding 24 h accumulation. This is due to the fact that not all 30-min periods may meet the criteria for inclusion in the 24 h event dataset. Application of a transfer function to periods that do not meet these event criteria is perfectly valid, but such an analysis is beyond the scope of this manuscript. In addition, the approach taken with the 12 and 24 h periods, whereby the presented transfer functions were

compared to transfer functions derived specifically for the 12 and 24 h periods, may not work well on time periods shorter than 30 min, as the amount of measurable solid precipitation will in many cases be approaching the noise level of the gauge.

(2) The authors present the complexity of errors at mountain stations. Especially the Weissfluhjoch station showed individual deviations at high precipitation – high wind speed events. Using a lower maximum wind speed thresholds results in smaller errors at this station. It will be advantageous if the authors present at least an advice on how to quality control data of mountain stations for such anomalies from the presented transfer functions without having a DFAR reference.

**Authors' response:** Unfortunately, without a DFAR or some other type of well-shielded reference, it is not possible to know how appropriate the presented transfer functions are for a given site. This issue was addressed in the Discussion (pg. 16. Ln. 1-10 in the originally-submitted manuscript). Here is an excerpt, "However, because one mountainous site was overcorrected (Weissfluhjoch), and the other two were undercorrected (Formigal and Haukeliseter), it is not possible to recommend general modifications to the transfer functions for use in mountainous sites." The advantage of the multi-site transfer functions and the multi-site evaluations in the present manuscript is that at least the uncertainty can now be estimated. The authors recommend using the transfer functions as presented, acknowledging the known uncertainties. This sentence has been added to the conclusions, "These results indicate that the transfer functions developed and presented here should be evaluated at new testbeds in the mountains and complex terrain, and also in other areas subject to high winds and unusual precipitation."

(3) Since no transfer function for adjusting liquid precipitation is presented, please consider to add "for mixed and solid precipitation" to the paper title.

**Authors' response:** The title has been changed to, "Analysis of single-Alter shielded and unshielded measurements of mixed and solid precipitation from WMO-SPICE". In an effort to make the title more concise, other words were removed.

Minor Comments

P3 Line 11ff: Please add the time interval of the data analysed.

**Authors' response:** This time period of the measurements is described in detail in the Methods section, but the phrase, "spanning two winter seasons from 2013-2015" has been added.

P3 Line 16: The last sentence may be shifted to the conclusion section.

**Authors' response:** The sentence has been moved to the first paragraph of the Conclusion.

P4 Line 21 and 23: Please present the expected min/max/range values for the 1 minute values.

**Authors' response:** This text refers to all of the WMO-SPICE measurements, including air temperature, wind speed, wind direction, and others, and as such includes many different min/max values. This has been clarified by changing the text from,

"A range check, to identify and remove points with values outside of the maximum and minimum expected values for each sensor", to "A range check, to identify and remove values that were outside of the manufacturer-specified output range for each sensor".

P5 Line 23: Please refer to the number of events given in Tab. 1 and 2.

**Authors' response:** Table 1 and 2 do not include all of the SEDS measurements, as they were subject to further quality assurance and threshold tests as described in the section titled, "Filtering of precipitation events". Because of this it would be inaccurate to reference them here.

P7 Line 1: Add the information of time interval for aggregation of 30 min.

**Authors' response:** Thank you! This is especially important now that the manuscript includes analysis of other time periods. The text has been changed accordingly.

P8 Line 12: The 30 minute minimum thresholds of SLEDS are quite low. Disaggregating these values to 60 % of 30 minutes results in precipitation rates of a minimum of 0.001 to 0.002mm/minute. How do these values correspond to the nominal accuracy of the precipitation gauge?

**Authors' response:** It is indeed true that the precipitation rate of the SLEDS is quite low. However weighing gauges output the total weight of accumulated water, so in theory extremely low-rate precipitation can be measured accurately if the time period is extended, allowing the gauge to accumulate a measurable change in weight. This is part of the reason accumulation periods shorter than 30 min were not evaluated, as snowfall is frequently associated with low precipitation rates. Based on the manufacturers specifications, the 600 mm Geonor T200B accuracy is 0.6 mm (0.1% full scale), and its sensitivity is 0.05 mm. The output from this gauge is a frequency however, so in practice the resolution is higher than 0.1 mm if it the output is averaged or summed over a longer time period. The Pluvio$^2$ stated accuracy is 0.1 mm, and its resolution is 0.01 mm. In practice the accuracy of these gauges were fairly similar, and were both typically less than 0.1 mm, as can be seen from Figure 9 in the original manuscript (Figure 10 in the revised manuscript).

P9 Line 16ff: Is it the maximum threshold of the 30 min average wind speed or is it the maximum wind speed in the 30 min interval. Please clarify.

**Authors' response:** This is an excellent question! We have clarified in the manuscript that the threshold was applied to the mean 30-min wind speed. To further clarify, the threshold is now referred to as the "wind speed threshold", rather than the "maximum wind speed" throughout the text.

P11 Line 25ff: The higher catch efficiency might also be caused by increased wind influence and thus undercatch at the DFAR. Please discuss.

**Authors' response:** Thank you, this is indeed possible. This has been added to the manuscript, "Heterogeneous winds and/or significant mean vertical wind speeds may also have caused the DFAR to underestimate precipitation in high winds."

P12 Line 15: Please define "other sites".

**Authors' response:** A list of the other sites has been added in parentheses, "(CARE, Sodankylä, Caribou Creek, Marshall, and Bratt's Lake)".

P12 Line 18: Replace "alpine measurements" by "measurements at mountain sites"

**Authors' response:** "Alpine" has been replaced by "mountainous" throughout the text.

P15 Line 29: (here and throughout the text) Are the same stations meant with "alpine" and "mountain" sites? If so, please consider to use only one of the two.

**Authors' response:** "Alpine" has been replaced by "mountainous" throughout the text.

Figures: Figure 2 and 6: To show the temperature dependence of Eq.3 please present additional calculations for at least one warmer and one colder temperature level ( e.g. -2 and -10∘C).

**Authors' response:** Due to the relative insensitivity of the transfer functions to temperature below -2 °C, the addition of these other cold temperatures do not add much value, and they detract from the main point of these figures. Figure B1 is included below as an example.

**Figure B1. Transfer functions describing the unshielded (UN) catch efficiency (*CE*) as a function of the gauge height wind speed ($U_{gh}$). The Eq. 3 results were produced by modelling *CE* with respect to wind speed at $T_{air}$ = -5 °C, and both the Kochendorfer et**

al. (2016) pre-SPICE (red line) and the current results (blue line) are shown. In addition, the Eq. 3 function is shown at $T_{air}$ = -2 °C and $T_{air}$ = -10 °C (dashed lines). The Eq. 4 snow ($T_{air}$ < -2 °C) results (green line) are also shown.

---

## Author Comment (AC4) · 22 May 2017

**Response to E. Lanzinger**

**General Comments:**

Large measuring uncertainties for solid precipitation measurements are a major issue in hydrology.

The paper „Errors and adjustments for single-Alter shielded and unshielded weighing gauge precipitation measurements

5   from WMO-SPICE" by John Kochendorfer et al. is clearly a substantial contribution to improved solid precipitation

measurements.

Besides improved, robust correction formulea there is also an estimate of the remaining uncertaitiy given. Outliers in the

results are clearly explained and hints for improvement are given.

The text is well stuctured and language, explanations, graphs and tables are very clear and don't need any change. I have

10  only minor questions in some parts That I address below.

The intention of this paper is to provide a simple and practical method for correcting solid precipitation data by ancillary data

that is available at most of the meteorological stations world wide.

The scientific significance, quality, and the presentation quality are therefore excellent.

**Authors' response:** Thank you very much!

15

**Specific Comments:**

P. 1, lines 23/24: Choose a more appropriate word for „differences". What you mean here are „errors" or „measuring

uncertainties". The word „difference" is too neutral, i.e. different amounts of precipitation could indeed occur in different

regions without being an error.

20  **Authors' response:** Thank you; "differences" has been replaced with "measurement biases" in the revised manuscript.

P. 2, line 4: In „...changes in the velocity of the air around the gauge…" velocity and direction of the airflow could be added,

as the air flow is bended around the gauge which is also leading to an uplift of light particles over the gauge orifice.

25  **Authors' response:** The word 'velocity' actually includes both speed and direction, but it has been replaced with "speed and

direction" for clarity, and the word "airflow" has been included.

P. 2, line 19/20: Could you eventually find the original citations that served as a rationale for the WMO decision of 2010?

Your citation is of 2012. you could use

30  **Authors' response:** This is a good point. Rasmussen et al. (2012) has been replaced with these: (Førland and Hanssen-

Bauer, 2000; Goodison et al., 1998; Sevruk et al., 2009).

P. 2, lines 27-29: It is not understandable for everybody, what the term „WMO-SPICE weighing precipitation gauges" means. I suggest two sentences: The focus of the work described below is on unshielded and single-Alter-shielded weighing precipitation gauges. Based on results of a previous CIMO survey (Nitu and Wong, 2010), WMO-SPICE selcted two weighing gauges which represent the two most ubiquitous configurations used in national networks for the measurement of solid precipitation.

**Authors' response:** The text has been changed accordingly.

P. 2, line 32: DFIR needs a citation.

**Authors' response:** An appropriate citation has been added (Goodison et al., 1998).

10

P. 3, lines 7-8: „Some of this variability is driven by differences in ice crystal shape (habit), mean hydrometeor fall velocity, and hydrometeor size...". Actually hydrometeor fall velocity is a function of its size, shape and mass (or mass density). Therefore it should not be mentioned side by side with the other variables. Hydrometeor shape and density are very difficult to measure for each particle, and are therefore generally not available. But some disdrometers provide hydrometeor size and fall velocity. By using fall velocity you implicitely take into account crystal shape and mass density. Please check again the given citation (Thériault et al. 2012) for details.

**Authors' response:** This list of attributes may admittedly seem redundant, but the mass and surface area of a hydrometeor affect not only its fall velocity, but also its inertial tendencies. Two hydrometeors with the same fall velocity may respond differently to the flow disturbance around a gauge due to differences in their mass. A larger hydrometeor will more readily cross local streamlines around the inlet due to its increased inertia. This effect is described by Colli et al. (2015).

P. 9, line 3: „mixed precipitation was defined as..."
I wonder whether „mixed precipitation" is a good term here, because you would expect a mixture of rain an snow for each of the events. I guess most of the events were snow only? Isn't the temperature used to distinguish between wet and dry snow? I would suggest „wet snow/mixed precipitation" and for the colder regime eventually „dry snow."

**Authors' response:** In this case temperature is being used to distinguish between precipitation types, because at monitoring sites where precipitation type is not measured, air temperature is the only available proxy for precipitation type observations. No differentiation between wet snow and dry snow is made in this manuscript, because below -2 °C the catch efficiency was relatively insensitive to air temperature (e.g. Figure B1 in the response to Helfricht's comment). The temperature region defined as mixed ($2 \,°C \geq T_{air} \geq -2 \,°C$) contains measurements that cannot with any degree of confidence be defined as either solid or liquid. It is indeed true that many of the individual 30-min events are not mixed precipitation, and are dominated by either rain or snow (eg. Kochendorfer et al., 2017; Wolff et al., 2015), but the set of 30-min measurements this temperature regime can justifiably be described as 'mixed' because it includes snow, mixed, and liquid precipitation. Wet snow/mixed precipitation would not describe it well, as it does include a significant amount of rain.

P.9, line 5: I suggest to delete the phrase „and the negligible magnitude of the liquid precipitation adjustment", because for drizzle the CE in windy conditions is negatively affected and needs correction. What you might have in mind is that the corrections for snow are generally larger. But as the fall speed of small droplets and small snow particles are very close it is

5   clear that both will be affected by wind in a comparable way.

**Authors' response:** It is indeed true that drizzle measurements can be underestimated due to wind, but within these WMO-SPICE measurements there was no significant undercatch of liquid precipitation. This is demonstrated by the similarity of the measurements adjusted using Eq. 3 and Eq. 4, as the liquid precipitation measurements were corrected using Eq. 3, and were uncorrected using Eq. 4. This has been clarified by re-writing as, "…the negligible magnitude of the liquid

10   precipitation adjustment derived from these measurements".

P. 13, line 1: PE was improved by the application of transfer functions for all sites, except one: Sodankylä. It is really a very very small difference, but it could be mentioned that in a site that is well shielded by trees and where there are generally low wind speeds, a SA shield is already sufficient and no further correction is needed. As you mentioned, the correction curves

15   are not precise at the low and high wind speed end. Could this be a reason, why the Sondakylä results with SA shield are getting a bit worse after correction? With the same argument, that you require to keep the correction constant for wind speeds higher than 7.2 m/s you could think about keeping the correction constant to 1 (or some value close to 1) for wind speeds lower than a threshold. By these two means you get closer to the sigmoid curves published elsewhere.

**Authors' response:** The following text has been added to the single-Alter results section, "At Sodankylä however, the

20   single-Alter shielded measurements were not significantly improved by the adjustments, and the $PE_{0.1\,mm}$ values were actually slightly lower after adjustment. This indicates that at a field site such as Sodankylä, which was well sheltered from the wind in a forest clearing, such adjustments may not be necessary." It may indeed be possible that the Sodankylä results were affected by inaccuracies in the adjustments at low wind speeds. Our hope was that the inclusion of the many low wind speed measurements from Sodankylä would minimize this problem in the resultant transfer function. Indeed the value of the

25   transfer function could be fixed at 1.0 near $U = 0$. The sigmoid function and the present Eq. 3 have been compared previously using a separate dataset from two sites, with no significant improvement to the sigmoid function detected (Kochendorfer et al., 2017). The two equations will not perform the same in all cases, and any equation is of course only as good as the data it has been fit to. However in the present case, based on the negligible differences between the adjusted and unadjusted measurements from Sodankylä, it does not appear that the added complexity of a lower wind speed threshold is

30   warranted. In addition, without performing a detailed analysis, comparison of the Eq. 3 and Eq. 4 adjustments shown in Fig. 6, with larger predicted CE for the Eq. 4 curve at low wind speeds, does not indicate that differences in the low wind speed adjustments had a significant effect on the Sodankylä $PE_{0.1\,mm}$ values (Fig. 7 d).

**Typos and linguistic comments:**

P. 5 line 11: rationale

**Authors' response:** A definition of 'rationale' is available here: https://www.merriam-webster.com/dictionary/rationale

P. 5, line 15/16: „false precipitation error" sounds a bit like „false error". I suggest „false accumulation" instead.

5 **Authors' response:** Thank you very much; "precipitation error" will be changed to, "accumulation".

P. 6, line 2: „10 m wind speed". I suggest „wind speed at 10 m height"

**Authors' response:** Thank you very much; "10 m wind speed" will be changed to, "10 m height wind speed".

10 P. 8, line 4: threshold of 0.1 mm in 30 min was chosen… the 30 min could be added.

**Authors' response:** Thank you very much; this will be changed accordingly.

P. 17, line 1: ...reduce the horizontal wind speed impacting...

**Authors' response:** Thank you. This phrase has been reworded as, "which is to reduce the horizontal wind speed inside the

15 shield".

**References**

Colli, M., Rasmussen, R., Thériault, J. M., Lanza, L. G., Baker, C. B., and Kochendorfer, J.: An improved trajectory model to evaluate the collection performance of snow gauges, Journal of Applied Meteorology and Climatology, 54, 1826-1836,

20 2015.

Førland, E. J. and Hanssen-Bauer, I.: Increased precipitation in the Norwegian Arctic: True or false?, Climatic Change, 46, 485-509, 2000.

Goodison, B., Louie, P., and Yang, D.: The WMO solid precipitation measurement intercomparison, World Meteorological Organization-Publications-WMO TD, 65-70, 1998.

25 Kochendorfer, J., Rasmussen, R., Wolff, M., Baker, B., Hall, M. E., Meyers, T., Landolt, S., Jachcik, A., Isaksen, K., Brækkan, R., and Leeper, R.: The quantification and correction of wind-induced precipitation measurement errors, Hydrol. Earth Syst. Sci., 21, 1973-1989, 2017.

Rasmussen, R., Baker, B., Kochendorfer, J., Meyers, T., Landolt, S., Fischer, A. P., Black, J., Theriault, J. M., Kucera, P., Gochis, D., Smith, C., Nitu, R., Hall, M., Ikeda, K., and Gutmann, E.: How Well Are We Measuring Snow: The

30 NOAA/FAA/NCAR Winter Precipitation Test Bed, Bulletin of the American Meteorological Society, 93, 811-829, 2012.

Sevruk, B., Ondras, M., and Chvila, B.: The WMO precipitation measurement intercomparisons, Atmospheric Research, 92, 376-380, 2009.

Wolff, M. A., Isaksen, K., Petersen-Overleir, A., Odemark, K., Reitan, T., and Braekkan, R.: Derivation of a new continuous adjustment function for correcting wind-induced loss of solid precipitation: results of a Norwegian field study, Hydrology and Earth System Sciences, 19, 951-967, 2015.

---

## Author Comment (AC5) · 22 May 2017

Because some of the changes made in response to the referee comments were fairly extensive, they have all been documented in a tracked-changes version of the manuscript available below.

Please also note the supplement to this comment:
http://www.hydrol-earth-syst-sci-discuss.net/hess-2016-684/hess-2016-684-AC5-supplement.pdf
* * *